



# Strong control of the stratocumulus-to-cumulus transition time by aerosol: analysis of the joint roles of several cloud-controlling factors using Gaussian process emulation

Rachel W. N. Sansom [1], Jill S. Johnson [2], Leighton A. Regayre [1,3,4], Lindsay A. Lee [5], and Ken S. Carslaw [1]

[1]School of Earth and Environment, University of Leeds, Leeds, UK
[2]School of Mathematical and Physical Sciences, University of Sheffield, Sheffield, UK
[3]Met Office Hadley Centre, Exeter, UK
[4]Centre for Environmental Modelling and Computation, University of Leeds, Leeds, UK
[5]Advanced Manufacturing Research Centre, University of Sheffield, Sheffield, UK

**Correspondence:** Rachel W. N. Sansom  (r.sansom@leeds.ac.uk)

**Abstract.** Stratocumulus-to-cumulus transitions are driven by numerous interacting cloud-controlling factors. Understanding these interactions is important for improving the accuracy of cloud responses to changes in climate and other environmental factors in global climate models. Many studies have found lower-tropospheric stability dictates the transition time, while aerosol-focused studies found that aerosol concentration plays a key role via the drizzle-depletion mechanism. We consider the role of aerosol together with several other cloud-controlling factors representing the wider environmental conditions. A 34-member perturbed parameter ensemble of large-eddy simulations with 2-moment cloud microphysics is used to train Gaussian process emulators (statistical representations) of the relationships between the factors and two properties of the transition: transition temporal length and average rain water path. Using these emulators, parameter space can be densely sampled to visualise the joint and individual effects of the factors on the transition properties. We find that in the low-aerosol regime ($< 200 \, \mathrm{cm}^{-3}$) the transition time is most strongly affected by the aerosol concentration. Fast transitions, under 40 hours, occur in this regime with high mean rain water path, which is consistent with a drizzle-depletion effect. In the high-aerosol regime, the inversion strength becomes more important than the aerosol concentration through the inversion's effect on entrainment and the deepening-warming decoupling mechanism.

## 1 Introduction

Stratocumulus-to-cumulus transitions occur in the east of major ocean basins when stratocumulus decks are advected towards the equator across increasingly warmer sea-surface temperatures (SST) (Klein and Hartmann, 1993; Albrecht et al., 1995). There is a large decrease in cloud fraction, albedo and cloud radiative effect as the cloud deck transitions to cumulus. The stratocumulus-to-cumulus transition is governed by many cloud-controlling factors, whose contributions are still an area of active research. Uncertain processes lead to poor parameterisations in global climate models so transitions are not captured well, which creates large uncertainties in simulated cloud properties and their responses to the warming climate (Bony and





Dufresne, 2005; Teixeira et al., 2011; Eastman et al., 2021). Low clouds in the subtropics have a cooling effect on the planet, so future decreases in cloud fraction will reduce that cooling effect, amplify warming, and contribute to a positive cloud feedback effect (Bretherton, 2015; Ceppi et al., 2017; Nuijens and Siebesma, 2019). Further process understanding of cloud transitions will improve their representation in global climate models and reduce the uncertainty surrounding cloud adjustments and
feedbacks.

The typical transition mechanism, termed deepening-warming decoupling, has been determined through observational studies (Paluch and Lenschow, 1991; Bretherton and Pincus, 1995; Bretherton et al., 1995; Martin et al., 1995; Wang and Lenschow, 1995; Klein et al., 1995; de Roode and Duynkerke, 1996; Pincus et al., 1997) and high-resolution modelling (Krueger et al., 1995; Wyant et al., 1997; Bretherton and Wyant, 1997; Svensson et al., 2000). It describes how increasing SSTs cause the
boundary layer turbulence to be increasingly driven by surface fluxes that deepen the boundary layer. As the boundary layer deepens, mixing throughout the full layer can no longer be sustained and the layer decouples into a stratocumulus cloud layer and a surface-coupled sub-cloud layer. Once decoupled, the stratocumulus layer is cut off from the ocean as a moisture source, but the sub-cloud layer becomes more turbulent, warmer and moister from surface evaporation until cumulus plumes develop. In this cumulus-under-stratocumulus stage, the plumes at first provide moisture and turbulence to the stratocumulus layer, but
more-energetic plumes overshoot and vigorous mixing eventually dissipates the stratocumulus cloud resulting in a field of cumulus.

The role of drizzle in the transition has historically been inconsistent between studies (Miller and Albrecht, 1995; Wang, 1993; Pincus et al., 1997; Svensson et al., 2000). Several modelling studies have found that drizzle plays a small role compared to other cloud-controlling factors (Sandu and Stevens, 2011; McGibbon and Bretherton, 2017; Blossey et al., 2021). For exam-
ple, Sandu and Stevens (2011) perturbed cloud-controlling factors in a large-eddy simulation (LES) of a composite case derived from thousands of trajectories in the North East Pacific (Sandu et al., 2010). Reducing cloud droplet number concentration from 100 to 33 $\text{cm}^{-3}$ allowed precipitation to form earlier and limited boundary layer recovery from decoupling through moistening and cooling the sub-cloud layer and depleting the cloud layer of water. The cloud did break up faster, but the initial strength of the temperature inversion capping the boundary layer had a stronger control on the timing of the breakup. However, as in many
LES studies, a fixed droplet number was used, while Yamaguchi et al. (2017) showed that aerosol collision-coalescence processes are required to represent droplet depletion. A recent study that used a detailed microphysics scheme Chun et al. (2025) found that including a large-scale circulation parameterisation overrides the effect of aerosol injection in lightly precipitating conditions.

Including collision-coalescence processes in LES models ensures there is a feedback between the reduction of droplets as
they collide and the reduction in aerosol number concentration, which then further reduces cloud droplet number. Using an LES model with a microphysics scheme that included this processing, Yamaguchi et al. (2017) found that a fast transition mechanism is initiated in a low-aerosol environment. They proposed that drizzle droplets are formed in cumulus plumes and strong updrafts carry them to the stratocumulus layer where they enhance drizzle production because they are larger than the stratocumulus cloud droplets, and therefore more efficient collectors. Through collision-coalescence and wet scavenging, the
droplet number and aerosol concentrations are reduced leading to even heavier drizzle, more reduction and a runaway feedback.





Using the same model for a different case, Diamond et al. (2022) also found a rapid reduction in cloud fraction through drizzle depletion in low aerosol conditions, with an end state closer to open-cellular organisation rather than cumulus. Erfani et al. (2022) used single-mode bulk microphysics that included aerosol processing within cloud droplets, and also found precipitation to be a key driver of the transition. These studies do not fully consider the effect of aerosol concentration in the context of other

cloud-controlling factors: Diamond et al. (2022) perturbs some large-scale forcings but with a focus on smoke effects, while the trajectories in Erfani et al. (2022) have very different initial conditions but cover only two extreme cases.

Observations from ships and satellites, along with reanalysis data, provide wider meteorological context (e.g. Mauger and Norris, 2010), but they have not shown clear evidence of a rapid transition to cumulus by a drizzle-depletion mechanism (Pincus et al., 1997; Zhou et al., 2015; Brendecke et al., 2021). Eastman and Wood (2016) analysed Lagrangian trajectories

from satellite data to study how boundary layer depth, the inversion strength and precipitation affect cloud evolution. Deep boundary layers and weak inversions tended more towards cloud breakup, but precipitation effects were less clear: in shallow boundary layers, precipitation sustained the cloud whereas in deep boundary layers it caused cloud breakup. Despite finding that increases in aerosol increased average cloud fraction, Christensen et al. (2020) also did not find precipitation or low aerosol to be a strong driver of cloud breakup. Eastman et al. (2022) assessed the difference between closed-cell stratocumulus that does

and does not transition. Heavy precipitation was linked closely with a transition to open-cell stratocumulus, but the transition to a cumulus state is more likely caused by excess entrainment at cloud top.

High-resolution model simulations of the transition have been limited to one-at-a-time perturbations, or only a few detailed trajectories, which sample only a few points in what is a multi-dimensional "parameter space" created by all the cloud-controlling factors. Sandu and Stevens (2011); Van Der Dussen et al. (2016); Zheng et al. (2021) made large one-at-a-time

perturbations to meteorological conditions, such as subsidence, droplet number, radiation and latent heat fluxes. LES model intercomparisons of the transition compared with observations highlight which structural differences create the largest disparities in replicating observed transitions (Bretherton et al., 1999; van der Dussen et al., 2013; de Roode et al., 2016). Small perturbations to initial conditions can represent different stages of the transition (Chung et al., 2012; Tsai and Wu, 2016; Bellon and Geoffroy, 2016), while simulating observed or calculated trajectories with completely different sets of initial conditions

produces very different transition characteristics (Goren et al., 2019; Blossey et al., 2021; Erfani et al., 2022). Within these studies, precipitation is found to have no effect or to slightly hasten the transition but it is not found to be a key driver. However, because these studies could only sample parameter space a few times, covariance between some meteorological factors may have been overlooked and so missing interactions between factors (Feingold et al., 2016).

Using Gaussian process emulation, a statistical representation (an emulator) can be created of the multi-dimensional rela-

tionship between a set of cloud-controlling factors (parameters) and a property of the transition (O'Hagan, 2006). Training data in the form of a perturbed parameter ensemble (PPE) spans the multi-dimensional parameter space providing sufficient information, even though it is sparsely sampled. The trained emulators can be used to predict cloud property values for any new combination of input values, allowing us to quantify the contributions from each factor to the variance in the property (Saltelli et al., 2000; Johnson et al., 2015; Wellmann et al., 2018, 2020). The multi-dimensional parameter space can be densely sam-

pled using the emulators to create response surfaces, which enable us to visualize non-linear joint effects of factors or the





relationships between cloud states, e.g., Glassmeier et al. (2019) and Hoffmann et al. (2020). Johnson et al. (2015) identified distinct behavioral regimes when analyzing their PPE of deep convection before emulating, and in Sansom et al. (2024) we used response surfaces to visualize regimes of stratocumulus cloud behavior.

In this study we have used an LES model to create an ensemble of stratocumulus-to-cumulus transitions initiated with a wide range of meteorological conditions covering key cloud-controlling factors. We define "transition" as the time (in hours) taken to transition from the initial stratocumulus state to a cumulus state. Given the potential importance of drizzle formation, the ensemble also samples a range of rain autoconversion rates. Each of these perturbed factors has the potential to affect the characteristics of the transition, and in perturbing them simultaneously and in various combinations, we can learn how they jointly affect the transition. We then apply Gaussian process emulation to the PPE to create emulators of transition time and average rain water path. We address the following questions. 1) What combination of factors is most important in determining the transition time? 2) What combination of factors is most important in determining the drizzle amount, and how does drizzle affect the transition time? 3) Under what conditions might a drizzle-depletion mechanism occur?

## 2 Simulation and ensemble design

### 2.1 Model configuration

The PPE is based on the composite case created for the NE Pacific Ocean basin in Sandu and Stevens (2011). Sandu et al. (2010) calculated thousands of forward and backward air parcel trajectories from areas of extensive cloud cover, over six days of advection, and retrieved the boundary layer properties for this period from satellite data and meteorological reanalysis. A reference case was designed from a subset of trajectories for the three days in which the majority of the transition occurred (Sandu and Stevens, 2011). The meteorological state in this reference case is a good starting point for simulating a typical transition in the NE Pacific, from which we perturbed a range of cloud-controlling factors to explore variations in cloud behaviour.

The ensemble was simulated using the UK Met Office and National Environmental Research Council (NERC) LES model, called the MONC (Met Office/NERC Cloud) model (Dearden et al., 2018; Poku et al., 2021; Böing et al., 2019). The model solves a set of Boussinesq-type equations, using an anelastic approximation here, which is based on a reference potential temperature profile that depends only on height. The subgrid turbulence parameterization is an extension of the Smagorinsky-Lilly model and is based on that described in Brown et al. (1994). Here, MONC was coupled to the two-moment Cloud AeroSol Interaction Microphysics scheme (CASIM) (Shipway and Hill, 2012; Hill et al., 2015) and the Suite of Radiation Transfer Codes based on Edwards and Slingo (SOCRATES) (Edwards and Slingo, 1996).

Stratocumulus-to-cumulus transitions are often simulated in a Lagrangian style in which the domain moves with the advected cloudy air (Krueger et al., 1995; Sandu and Stevens, 2011; de Roode et al., 2016). As in other studies, we simulated the advection towards the equator by forcing the SSTs to increase over the course of the simulation. Wind profiles were retained to ensure appropriate ocean surface evaporation, but the model has periodic boundary conditions so the domain was always focused on the same cloud cell. Simulations were run for 3-4 days and SST increased by nearly 1.5 K per day, following Sandu





**Table 1.** Parameter descriptions, symbols and ranges in parameter space.

| Parameter description | Symbol | Range |
|---|---|---|
| Boundary layer vapor mass mixing ratio | BL $q_{\mathrm{v}}$ | 7 to 11 g kg$^{-1}$ |
| Boundary layer depth | BL $z$ | 500 to 1300 m |
| Inversion jump in potential temperature | $\Delta\theta$ | 2 to 21 K |
| Inversion jump in vapor mass mixing ratio | $\Delta q_{\mathrm{v}}$ | -7 to -1 g kg$^{-1}$ |
| Boundary layer aerosol concentration | BL $N_{\mathrm{a}}$ | 10 to 500 cm$^{-3}$ |
| Autoconversion rate parameter (Khairoutdinov and Kogan, 2000) | $b_{\mathrm{aut}}$ | -2.3 to -1.3 |

and Stevens (2011), Bretherton and Blossey (2014) and Yamaguchi et al. (2017). The domain was 12.8 by 12.8 by 3.1 km$^3$. The

horizontal resolution was 50 m, and the vertical resolution varied from 20 m near the surface, to 5 m around the temperature
inversion, and gradually increased above that.

CASIM is a two-moment bulk microphysics scheme that represents hydrometeors using gamma distributions for mass and
number (Grosvenor et al., 2017). Only warm-cloud processes (cloud liquid and rain) were used since ice processes are not
part of the stratocumulus-to-cumulus transition in the NE Pacific. Simulations were initiated with soluble aerosol represented

by prognostic mass and number concentrations in the Aitken and accumulation modes. The Aitken mode distribution has
a standard deviation of 1.25 and a mean radius of 25 nm. The accumulation mode distribution has a standard deviation of
1.5 and a mean radius of 100 nm. The density of all aerosol particles was assumed to be 1500 kg m$^{-3}$. At saturation, the
number of aerosol particles activated into cloud droplets was calculated using the scheme of Abdul-Razzak and Ghan (2000),
and these activated aerosol were represented using a separate in-cloud aerosol prognostic. Aerosol material contained within

droplets can grow through droplet collision and coalescence with the assumption that one aerosol particle was present in each
droplet, and is returned to the appropriate aerosol size mode on evaporation of the cloud droplets (including the coarse mode).
Accretion and autoconversion are represented by the Khairoutdinov and Kogan (2000) parameterization. Rain can evaporate in
the subsaturated grid boxes, but aerosol is not returned to the size modes through this process.

## 2.2    Perturbed parameter ensemble

PPEs are a valuable tool for understanding the joint effects of parameters on model output. Perturbing parameters simultane-
ously in a space-filling way maximizes information from the model about how parameters jointly affect the outputs of interest.
Five cloud-controlling factors were perturbed plus a sixth factor that alters the dependence of the autoconversion rate on $N_d$.
Table 1 shows the individual ranges for each parameter, which form the boundaries of the 6-dimensional hypercube that the
ensemble covers. The following paragraphs describe each parameter and their roles in cloud transitions.



### Boundary layer vapor mass mixing ratio

The boundary layer vapor mass mixing ratio (specific humidity) directly determines at what point saturation is reached and how much moisture is available for cloud droplets to form. It also determines how much drizzle will be evaporated below cloud base.

### Inversion properties

The strength of the inversion was perturbed by two properties: the jump in potential temperature and specific humidity across the inversion. The dissipation of the stratocumulus cloud is a defining feature of the transition and is largely caused by the entrainment of warm, dry air from above the inversion, via overshooting cumulus plumes. Thus, the rapidity of this dissipation is related to the strength of the inversion and the specific humidity in the free troposphere (Wood et al., 2018), which can be perturbed with the changes in temperature and moisture across the inversion (the jump in potential temperature will be used interchangeably with inversion strength). Additionally, the free-tropospheric humidity determines the rate of longwave cooling, which affects entrainment and evaporation (Siems et al., 1993).

### Boundary layer depth

The boundary layer depth determines how well the layer can mix and consequently how well supplied with surface-evaporated moisture the stratocumulus cloud layer is. Eastman and Wood (2016) showed that precipitation may have opposite effects on stratocumulus cloud transitions depending on whether it is occurring in deep layers, leading to break up, or shallow layers, leading to cloud persistence.

### Boundary layer aerosol

The initial boundary layer concentration of accumulation mode aerosol was perturbed because the vast majority of aerosols that activate into cloud droplets (cloud-condensation nuclei) are from the accumulation mode. Free-tropospheric aerosol can also be a source of cloud-condensation nuclei and could be important in simulations with very low aerosol concentrations in the boundary layer (Wyant et al., 2022). However, free-tropospheric aerosol concentration was kept constant across the PPE because it was not expected to be as important as the key factors chosen.

### Autoconversion rate parameter

The autoconversion rate determines how readily cloud droplets form rain droplets in a parameterisation of the collision-coalescence process. In the Khairoutdinov and Kogan (2000) parameterisation, the autoconversion rate is given by

$$\left(\frac{\delta q_{\mathrm{r}}}{\delta t}\right)_{\mathrm{auto}} = 1350 q_{\mathrm{c}}^{2.47} N_{\mathrm{d}}^{-1.79},$$





where $q_r$ is the rain mass-mixing ratio, $q_c$ is the cloud liquid mass-mixing ratio (both in kg kg$^{-1}$), and $N_d$ is the cloud droplet number concentration (cm$^{-3}$). We perturbed the exponent of cloud droplet number concentration from the default value of -1.79. The default parameter values were estimated in Khairoutdinov and Kogan (2000) by reducing the mean squared error between the above function and an explicit microphysics model, and there are large uncertainties surrounding each of these values.

### 2.2.1 Perturbation method

The perturbation values were chosen using a "maximin" Latin hypercube approach. Figure 1 shows the 6-dimensional design, which maximizes the minimum distance between points to ensure that values are well-spaced across the multi-dimensional parameter space and each 1-dimensional axis (Morris and Mitchell, 1995). The values for the autoconversion parameter have been transformed using the inverse log because it is the exponent of $N_d$, i.e., the resulting autoconversion rates were approximately uniformly distributed, rather than the parameter values. The inset of Fig. 1 shows how these values in parameter space translate to initial conditions in the idealized model set up. The perturbed cloud-controlling factors shifted slightly during model spinup, but remained well spaced, so we analyzed the relationships between the post spin-up values and the transition properties.

We ran 85 simulations initially, but found that 31 did not form stratocumulus because the boundary layer was too shallow and dry. Out of those simulations that had stratocumulus, 26 did not transition to a cumulus state before the end of the simulation. It is unsurprising that not all of the simulations produced transitions because the initial conditions were broadly perturbed to sample a wide range of model behavior and not all parts of the joint parameter space are expected to be realistic. The remaining 28 simulations that transitioned to cumulus were augmented by 6 transitioning simulations, out of 12 points that were augmented to the original design. These points were augmented based on our new understanding of the regions of parameter space that produced stratocumulus and were likely to transition within simulation time, so increasing the density of information in the most relevant part of parameter space.

The parameter ranges were chosen to span the breadth of studies on stratocumulus-to-cumulus transitions in the subtropics. Often case studies are designed for LES simulation from observations of particularly fast or slow transitions, so a broad range of behaviours was included in the parameter space by spanning these reported cases (Sandu and Stevens, 2011; de Roode et al., 2016; Blossey et al., 2021). Since many LES studies have not focused on the aerosol effect, the range for the accumulation mode concentrations was informed by the Cloud System Evolution in the Trades (CSET) and Marine ARM GPCI Investigation of Clouds (MAGIC) campaigns (Bretherton et al., 2019; Painemal et al., 2015). Note that we have not included extremely polluted cases, such as the biomass burning region off the western coast of Africa. There are many studies of the aerosol semi-direct effect on the stratocumulus-to-cumulus transition in the Atlantic ocean, with some contradicting results (Yamaguchi et al., 2015; Zhou et al., 2017; Diamond et al., 2022). Further understanding of transition mechanisms will help to untangle these joint effects.

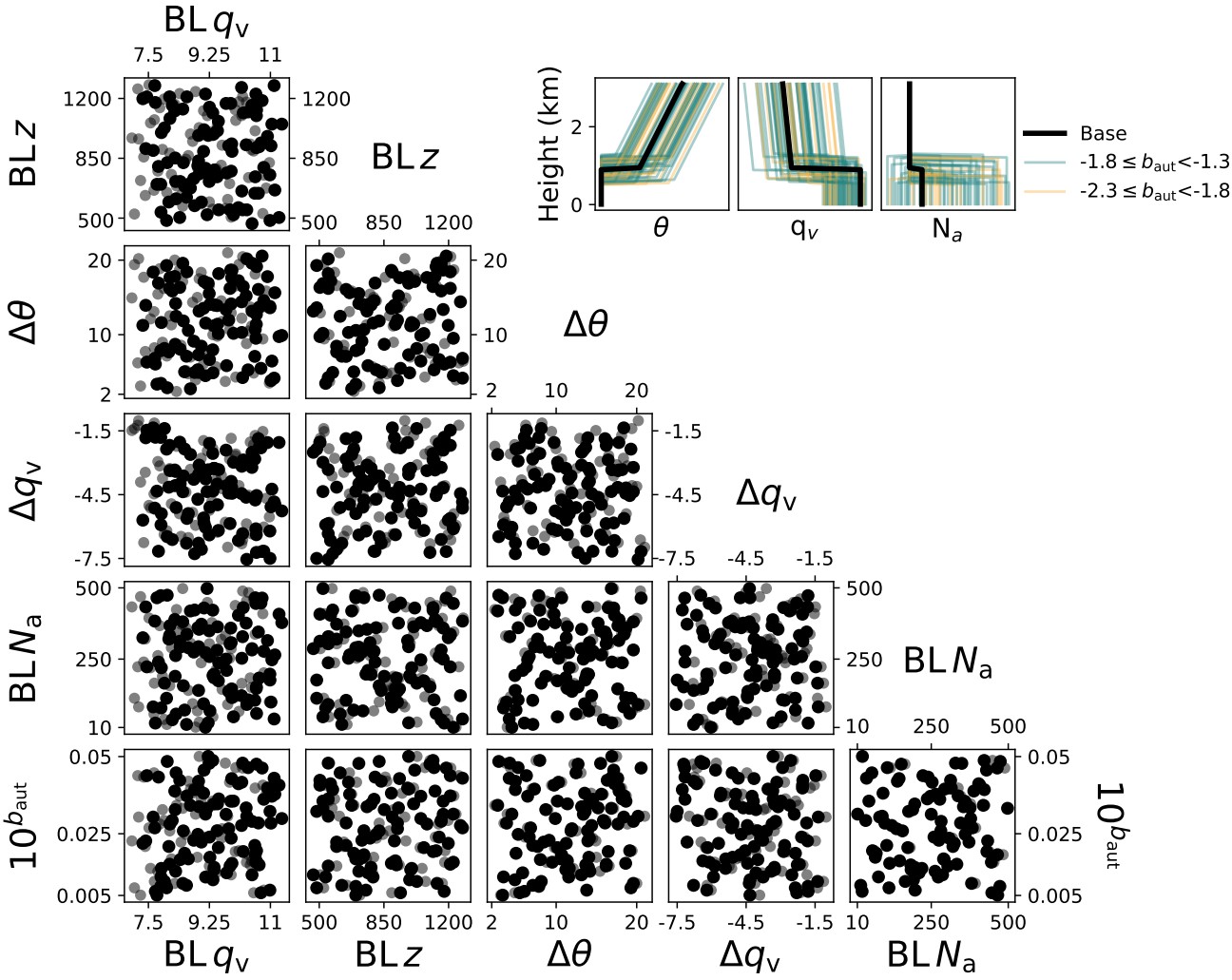

**Figure 1.** The Latin hypercube design for the perturbed parameter ensemble. Each 2-dimensional plot shows a different combination of two of the six parameters over the chosen ranges (see Table 1). The grey points show the values used for the initial conditions in each simulation from the original Latin hypercube design and the black points show how these values shifted after the model had finished spinning up. The inset shows how the parameters are perturbed in the initial profiles using this design.



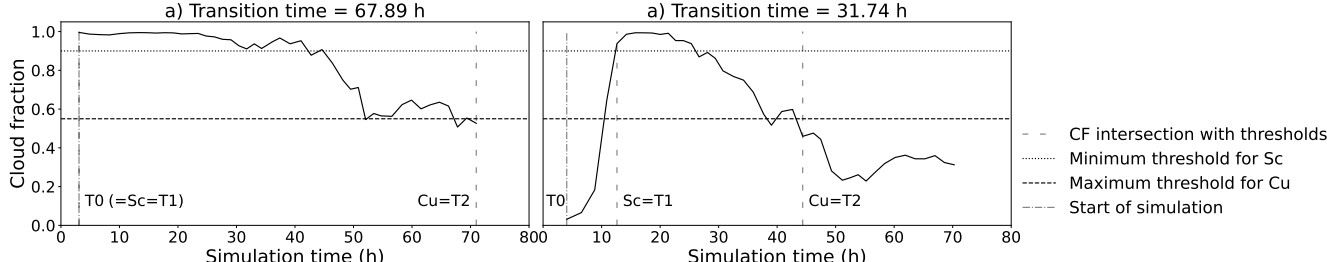

**Figure 2.** Transition time calculation based on cloud fraction. a) shows an ensemble member that has stratocumulus from the start of the simulation. b) shows a member that takes about 12 hours to build stratocumulus. The solid black line is the cloud fraction timeseries, the dotted line is the 0.9 threshold which is the minimum for stratocumulus, the dashed line is the 0.55 threshold which is the maximum for cumulus. The loosely dashed lines is where the cloud fraction intersects with the stratocumulus (Sc) and cumulus (Cu) thresholds.

## 2.3 Transition properties

The transition properties analysed here are the transition time and the mean rain water path ($R$). The transition time is the time taken to transition from a stratocumulus regime (beginning at T1) to a cumulus regime (beginning at T2). Figure 2 shows two examples of how this was calculated from the cloud fraction ($f_c$) for all the ensemble members based on $f_c > 0.9$ for stratocumulus and $f_c < 0.55$ for cumulus. Figure 2a shows the base simulation, which has stratocumulus from the start of the simulation (T0) so T1 is set equal to T0, although realistically T1 could be earlier. The $f_c$ decreases below the cumulus threshold just after 50 hours, but it recovers until the final time step when it reaches the threshold again, T2, giving a transition time of about 68 hours. It is possible the cloud could recover again if a longer simulation were conducted, which creates some noise in the calculation of transition time. Figure 2b shows a simulation that takes about 12 hours to build up stratocumulus, hence subtracting T2 from T1 gives a transition time of about 32 hours.

## 2.4 Gaussian process emulation

Gaussian process emulation is a Bayesian machine learning method to learn the relationship between a set of input parameters and an output of interest (Rasmussen and Williams, 2006; O'Hagan, 2006). It uses a prior specification of the relationship consisting of a mean function (e.g., constant or linear) and a covariance structure. Here we use a linear mean function and the Matérn 5/2 covariance structure. The prior is updated using a set of training data, which is the set of perturbed inputs and corresponding outputs from the PPE, to create a posterior specification. Once validated, the emulator can be used to predict values for new sets of input values, with quantified accuracy.

The emulators of transition time and mean $R$ were validated using the leave-one-out method. Here, an emulator is created from all but one of the training points and then used to predict a value for that left-out point. This is repeated for each point in the training set and the differences between the predicted values and the actual values are used to gauge how reliably the emulator can reproduce model output. Figure 3 shows that the training points were predicted within the 95% confidence intervals for all





but one of the points (97%) for transition time and mean $R$. However, the confidence intervals are quite large, especially in the transition time where some points are up to 10 hours out in the predictions. The mean $R$ emulates better because it is easier to
quantify than the transition time. There is some noise in the transition time calculation due to the simulation sometimes ending before it is obvious that the cloud has fully transitioned. The noise incurred in the transition time calculation is discussed in Section 4. We additionally validated the emulators by calculating the ratio of the standard deviation of the mean values at the training data (a measure of variation in emulated output) to the mean of the standard deviation of those points (the uncertainty in emulated values). For both emulators, this ratio is larger than 1, which tells us the function changes more than the underlying
emulator uncertainty. If the ratio was less than 1, the emulator uncertainty would be too large compared to changes in the function, so it would not be a useful approximation of the relationship. This validation shows that the emulators predict model output with sufficient accuracy for us to gain important insights into the processes that drive transitions.

Following our previous work in Sansom et al. (2024), we ran some initial condition ensembles to gauge the internal variability of the model, so that a "nugget" term could be added to the emulators. The nugget term allows the posterior mean function
to have a buffer around each training point, rather than interpolating them exactly. This is useful when the data are noisy or, as in this case, for incorporating internal variability. At four training points we ran four extra simulations and varied the random seed in the model that initiates turbulence, which allowed us to calculate the approximate variance due to internal variability in the transition time and mean $R$. In three of these initial-condition ensembles, the members all transitioned within a few hours of each other, but in one ensemble the cloud recovered and did not fully transition until early the next day (approx. 10 hours
later). Adding this variance into the emulators accounts for some of the noise created in the transition time calculation (Section 2.3). Details of this calculation are not shown here but may be found in Sansom et al. (2024) and in the code repository.

## 2.5 Variance-based sensitivity analysis

We used a Python package to calculate the Sobol indices, which obtain the contributions of variance in each parameter to the variance in the outputs that we are emulating (Sobol, 2001). We discuss the "main effect", which is how much of the variance
in the output is due to the variance in the individual parameter, and the interactions which are the portion of the variance that cannot be explained by linear combinations of the individual parameters, and is attributed to the interactions between parameters.

## 3 Results

We begin by evaluating the cloud properties in the base simulation (Section 3.1), which is central to our PPE design. We then
discuss the $f_c$ timeseries across the ensemble (Section 3.2), before assessing the controls on transition time (Section 3.3) and drizzle (Section 3.4) using the emulators.




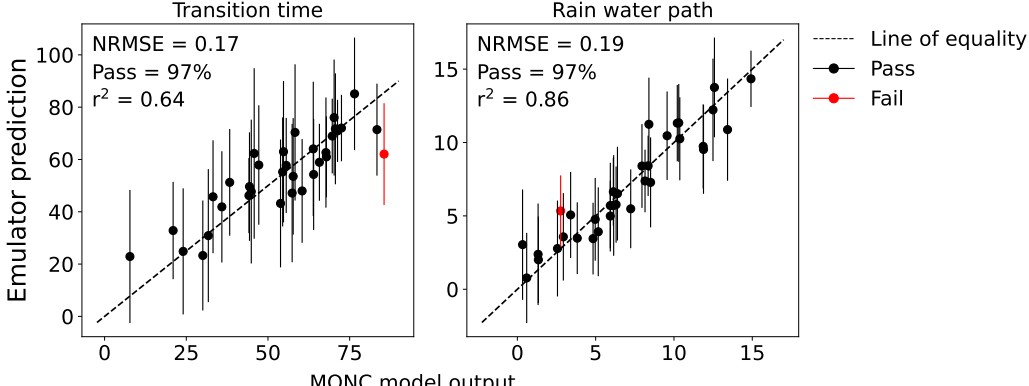

**Figure 3.** Emulator validation using the leave-one-out approach. Transition time is on the left and rain water path on the right. Points show the model output against the emulator-predicted values for each training data point that has been left out of the emulator training set in turn. Lines show the upper and lower 95% confidence bounds. Black points are where the model output data lies within the confidence bounds (pass) and red points are where this is not the case (fail).

## 3.1 Cloud properties in the base simulation

The stratocumulus-to-cumulus transition in the base simulation is similar to that of previous LES studies based on the Sandu and Stevens (2011) composite case (Bretherton and Blossey, 2014; Yamaguchi et al., 2017). Figure 4 shows three snapshots
of liquid water content from the beginning, middle and end of the simulation, and their associated $f_c$. The $f_c$ is defined as the fraction of cloudy columns with a cloud liquid mass-mixing ratio greater than 0.01 g kg$^{-1}$. Column a is from around 12 hours into the simulation and shows a uniform stratocumulus cloud with $f_c = 0.99$. The inversion height, and cloud top, are around 1000 m with a cloud layer thickness of about 300 m. Column b is from a day later and shows a slightly more broken cloud but still a high $f_c$ of 0.94. The cross section shows that the boundary layer deepened and cloud top rose by a couple of
hundred metres during the intervening day. The lowest cloud base remains around 700 m, but now the base marks the bottom of cumulus-like plumes that feed into the higher stratocumulus cloud base, around 100 m above. Since the first day, liquid water path ($L$) has decreased towards the edges of the cloud as the stratocumulus layer thinned. Column c is from the last two hours of simulation, at the end of the third day, and shows a much more broken cloud that is representative of a cumulus cell, with $f_c = 0.53$. At this stage the boundary layer is around another 100 m deeper and the cloud top has risen with it.
Compared to other studies that simulated this composite case, the boundary layer did not deepen to the same degree and there was less drizzle. Other LES models simulated a boundary layer depth between 1.5 to 2.5 km, whereas our simulation has a maximum depth of 1.4 km (Sandu and Stevens, 2011; Bretherton and Blossey, 2014; de Roode et al., 2016; Yamaguchi et al., 2017). This could be due to the different radiation schemes and mixing processes in the models, or to the stretching of the vertical layers in the top of the domain. In our simulation, $R$ peaks at about 25 g m$^{-2}$ at the beginning of the third
day, which aligns roughly with the sensitivity tests in Yamaguchi et al. (2017), which also used the Khairoutdinov and Kogan



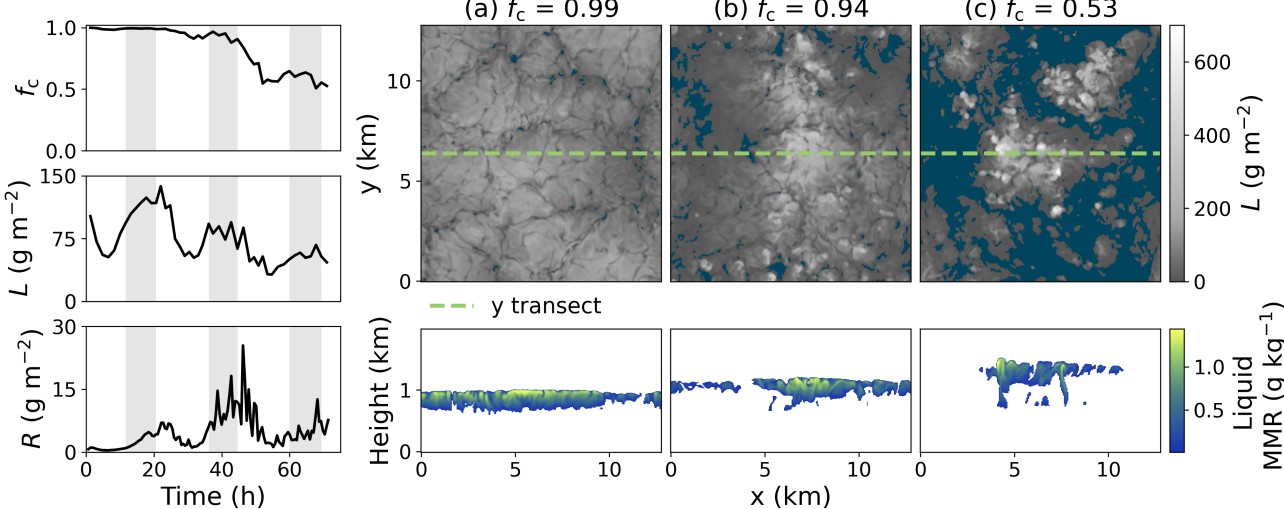

**Figure 4.** Base simulation cloud properties. Left: timeseries of cloud fraction ($f_c$, top), liquid water path ($L$, middle), and rain water path ($R$, bottom). The grey shading indicates local nighttime. Right: Snapshots of liquid water a) near the beginning of the simulation, ~12 hours, b) near the middle, ~34 hours, and c) at the end, ~72 hours. The top row shows the top-down view of liquid water path and the bottom row shows a vertical cross section of liquid water mass-mixing ratio (MMR) at the y-location of the transect line.

(2000) parameterisation in a similar domain size. However, it is much less than the peak of 150 g m$^{-2}$ for the same domain size using their bin-emulating bulk microphysics scheme. The transitions in our simulations may be slower than those in the previous studies because the shallower boundary layer may limit the boundary layer decoupling and the lower $R$ may limit the potential for a drizzle-depletion mechanism.

## 3.2 PPE cloud fraction analysis

Figure 5a shows that the range of initial $f_c$ produced across the PPE is large, as expected from perturbing many initial conditions over a large range of environmental conditions. Those that form stratocumulus (67 simulations, Fig. 5b) and those that form cumulus (37 simulations, Fig. 5c) make up the ensemble subset that transition. The subset mean in Fig. 5c is a similar shape to the base simulation, but the PPE members show a wide range of behaviours. On average, $f_c$ stays near one through the first day and night, before dipping in the second day to $f_c \approx 0.75$ and on the third day it crosses the cumulus threshold and stays below. A diurnal cycle can be seen in many of the members, with some members dipping to $f_c \approx 0.4$ and still recovering in the second night. Additionally, some members keep $f_c \approx 1$ until the third day and then transition rapidly.

While many of the simulations that transitioned formed stratocumulus within the first day, there were three simulations that only formed stratocumulus beyond the end of the second day when the SST had increased by at least 1 K and these transitioned very quickly. The transitioning simulations are "epoch aligned" in Fig. 5d by aligning T1 for each member, and the high SST members are highlighted. These fast transitions occur despite being in areas of parameter space where you might not expect





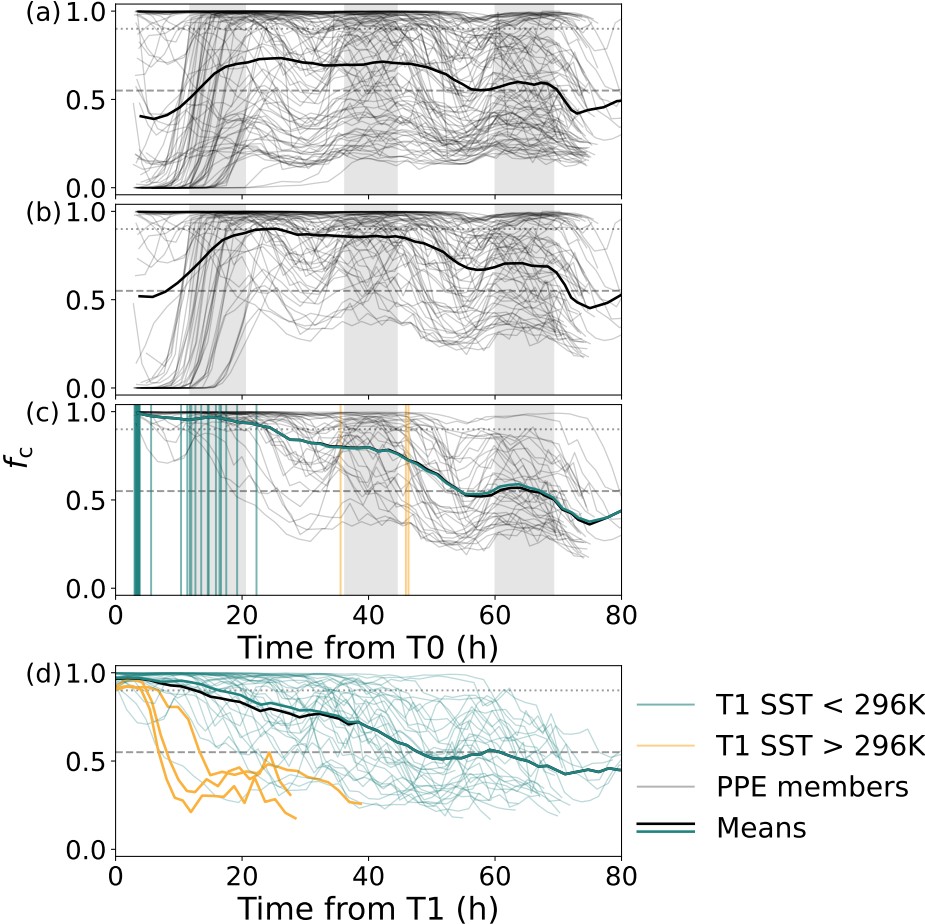

**Figure 5.** Cloud fraction timeseries for a) the whole ensemble, b) the members that form stratocumulus, c) the members that also form cumulus, and d) as in c) but aligned to the start of stratocumulus, T1. The thick, solid, black lines show the mean of the timeseries. The vertical lines show the start time of stratocumulus for each member, coloured either green or yellow depending on whether the SST at the start of stratocumulus is below or above 296 K. The ensemble members in d) are coloured by the SST threshold as well. The solid, green line shows the mean of the ensemble without the members with stratocumulus start SST above 296 K.



it, for example in a very shallow boundary layer with a low autoconversion rate. This subset of simulations shows that warmer initial SSTs may act to considerably speed up the transition, above meteorological conditions, which has implications for the future warmer climate. However, here these simulations have been removed from this analysis (leaving 34 simulations) since the difference in SST at initial stratocumulus is akin to perturbing a seventh parameter, but one that was not initially accounted for in our experimental design.

### 3.3 Transition time analysis

#### 3.3.1 Space-filling predictions

The emulator's posterior mean response surface was used to make 1000 predictions of transition time, which fill the parameter space and provide far more information than the raw PPE data alone. Figure 6 immediately begins to inform us about the subtleties in variation across parameter space. Some of the 2-dimensional subplots show clear variations, which means the transition time varies consistently for those two parameters over all values of the other parameters that are not shown in that panel (e.g., panels k and o). Other subplots show less clear variations of the transition time for the two parameters, which suggests there is no obvious dependence on these two parameters, or the effects of the four hidden parameters are dominating (e.g., panels a and c). There is a strong variation in transition time over the boundary layer aerosol concentration range, BL $N_\mathrm{a}$, with low BL $N_\mathrm{a}$ producing the fastest transitions (panels d, h, k, m, o). The inversion strength $\Delta\theta$ in panels b, f, j, k, l) and the autoconversion parameter ($10^{b_{aut}}$ along the bottom row) also cause strong variations in the transition time, which are particularly clear in combination with BL $N_\mathrm{a}$ (panels k and o).

#### 3.3.2 Transition time average response surfaces

The strength of the output's dependency on each parameter and the joint effects of parameters can be more easily interpreted in the averaged 1 million-point response surfaces in Fig. 7. The transition time has the strongest dependencies on aerosol concentration (BL $N_\mathrm{a}$), inversion strength $\Delta\theta$), and the autoconversion parameter ($b_{\mathrm{aut}}$). Most of the panels show at least linear joint effects (e.g., panels g and j) and several show non-linear joint effects (or interactions, shown by curved surfaces e.g., panels f, k, o) between parameters. Here we discuss the dependencies visualised in the response surfaces in Fig. 7 and suggest mechanisms from relevant studies.

The transition time has the strongest dependency on aerosol concentration, BL $N_\mathrm{a}$, (see panels d, h, k, m, o) with the fastest transitions corresponding strongly to stratocumulus in environments with low aerosol concentrations. The transition time is only predicted, on average, to be below 40 hours for BL $N_\mathrm{a}$ below 200 $\mathrm{cm}^{-3}$. There are clear joint effects with BL $z$, $\Delta\theta$, $\Delta q_\mathrm{v}$ and $b_{\mathrm{aut}}$ (panels h, k, m, o). Yamaguchi et al. (2017) and Diamond et al. (2022) found that low aerosol environments caused drizzle depletion of moisture and aerosol in the boundary layer. The deeper analysis in Yamaguchi et al. (2017) found that in their simulations it was specifically cumulus drizzle being lifted to the stratocumulus layer and initiating a rapid depletion. Erfani et al. (2022) found that adding aerosol into a clean case caused a delay in the transition, but adding aerosol into a polluted case had little effect on the transition time.



**Figure 6.** Transition time emulator sampled with a 1000-point Latin hypercube. a-o) shows each 2-dimensional combination of the six factors perturbed in the ensemble across the chosen ranges. The inset in the top right shows the contribution of each parameter's variance to the variance in the transition time.



The next strongest dependency is on the inversion strength, $\Delta\theta$ (panels b, f, j, k, l). The fastest transitions occur for stratocu-
mulus under weak inversions (small $\Delta\theta$) and the slowest transitions occur under strong inversions (large $\Delta\theta$). There are clear
joint effects with BL $z$, BL $N_\mathrm{a}$ and $b_\mathrm{aut}$ (panels f, k, l). Several studies have found the inversion strength, or the closely related
lower tropospheric stability, to be a key control on the transition time (Mauger and Norris, 2010; Sandu and Stevens, 2011;
Eastman and Wood, 2016). These studies showed that clouds under weak inversions are prone to break up or that clouds under
strong inversions persist. Strong inversions can trap moisture in the boundary layer and reduce boundary layer deepening and
decoupling, which is a key stage in the classic transition.

The third strongest dependency is on the autoconversion parameter, shown here as $10^{b_\mathrm{aut}}$ to be uniformly spaced (panels e, i,
l, n, o). The fastest transitions occur for high autoconversion rates. There are joint effects with BL $z$, $\Delta\theta$ and BL $N_\mathrm{a}$ (panels i, l
and o). Higher autoconversion rates would induce a drizzle-depletion effect as already discussed. In addition to the previously
mentioned studies, Eastman and Wood (2016) found a small, non-linear effect where precipitation sustains cloud cover in
shallow boundary layers but promotes cloud breakup in deep boundary layers. The interaction with BL $z$ in panel i agrees with
their suggestion that in the shallow case, precipitation creates stronger overturning circulation, but in the deeper case it deprives
the stratocumulus layer of moisture when it is also cut off from the ocean source.

The boundary layer depth, BL $z$, shows that stratocumulus in deep boundary layers on average transition faster than in
shallow boundary layer (panels a, f, g, h, i). The slight dependency of transition time on BL $z$ is seen more clearly in the joint
effects with $\Delta\theta$, BL $N_\mathrm{a}$ and $b_\mathrm{aut}$ (panels f, h, i). Wood and Bretherton (2006) showed that deep boundary layers are more
likely to be decoupled and, since decoupling is part of the classic transition mechanism, this stage could be accelerated when
beginning in a deeper boundary layer. Eastman and Wood (2016) found that clouds in deep boundary layers are prone to break
up, and they also suggested the transition is through decoupling.

The transition time is very weakly dependent on the jump in specific humidity, $\Delta q_\mathrm{v}$. The fastest transitions occur when there
is dry air above the boundary layer (panels c, g, j, m, n). Zhou et al. (2015) found that the entrainment of dry warm air at cloud
top was a major driver of decoupling through sudden drying of the boundary layer and subsequent rising of the condensation
point. Eastman et al. (2017) also found this pattern but suggested that more vapour above the cloud increases the downwelling
longwave, which offsets some of the longwave cooling, reducing mixing and boundary layer deepening. However, Sandu and
Stevens (2011) found that transitions were faster for increased downwelling longwave radiation. Our results suggest the strong
relationship that Zhou et al. (2015) and Eastman et al. (2017) found could be buffered by this effect.

The transition time is nearly invariant to changes in boundary layer specific humidity, BL $q_\mathrm{v}$, for any conditions of the
other parameters (panels a to e). There is a very weak relationship showing that more humidity in the boundary layer results in
longer transitions and vice versa. Moist boundary layers allow thicker clouds to form, which would then take longer to dissipate
through entrainment (Zhou and Bretherton, 2019).

The transition time sensitivity analysis, shown in top right of Fig. 7, quantifies the effects described above in terms of the
main effects (the average effect of a factor across all values of the other factors) and interactions. On average, the BL $N_\mathrm{a}$
main effect has the largest contribution to the variance in the transition time of 61%. The average $b_\mathrm{aut}$ main effect contributes
14%, $\Delta\theta$ contributes 9%, BL $z$ contributes 7%, $\Delta q_\mathrm{v}$ contributes 2% and BL $q_\mathrm{v}$ contributes less than 1%. The interactions from







**Figure 7.** Transition time response surface sampled with a 1-million point 6-dimensional grid and averaged across hidden dimensions. a-o) shows each 2-dimensional combination of the six perturbed factors averaged in the four dimensions not shown. The inset in the top right shows the contribution of each parameter's variance to the variance in the rain water path.





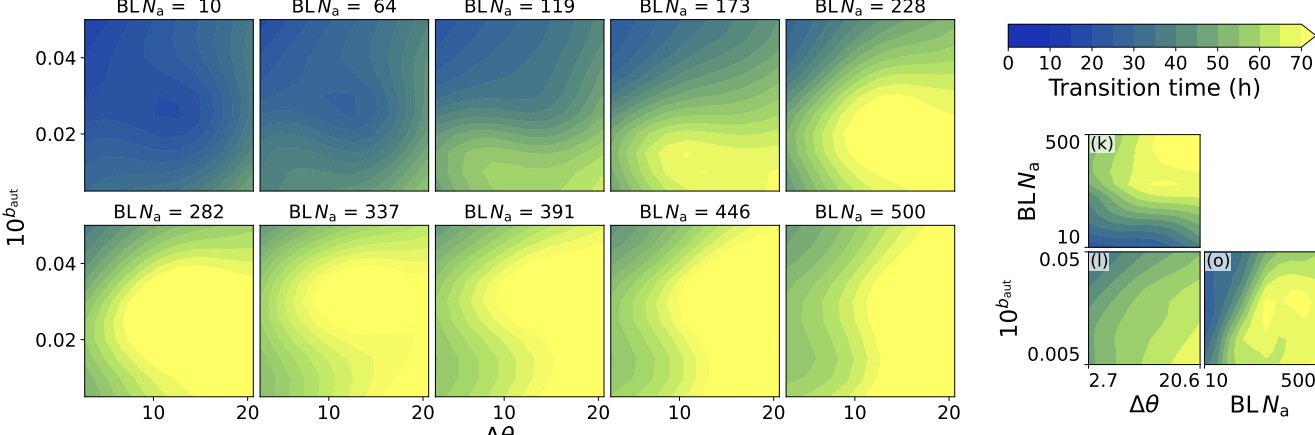

**Figure 8.** Transition time response surface slices. Averaged response surfaces for 10 values of $\mathrm{BL}\,N_\mathrm{a}$ in the $b_\mathrm{aut}$ and $\Delta\theta$ parameter space, averaged over the three parameters not shown. k, l, o) show the relevant averaged 2-dimensional plots from Fig. 7. k) $\Delta\theta$ vs $\mathrm{BL}\,N_\mathrm{a}$, l) $\Delta\theta$ vs $b_\mathrm{aut}$, which is the mean of the 10 plots shown here, and o) $\mathrm{BL}\,N_\mathrm{a}$ vs $b_\mathrm{aut}$.

each parameter contribute a total of around 7% of the variance, so the total interactions are more important than some of the
parameter main effects. The dependence on the interactions between parameters demonstrates the complexity of the transition
time drivers that more traditional studies have not managed to capture.

The response surfaces are visualised further in the three most important dimensions in Fig. 8: $\mathrm{BL}\,N_\mathrm{a}$, $\Delta\theta$, and $b_\mathrm{aut}$. The 10
slices at different $\mathrm{BL}\,N_\mathrm{a}$ values give a 3-dimensional picture of how transition time varies with all three parameters. Panels k,
l and o are the same as in Fig. 7. Panel l is the average of the 10 panels and panels k and o show $\mathrm{BL}\,N_\mathrm{a}$ with $\Delta\theta$ and $\mathrm{BL}\,N_\mathrm{a}$
with $b_\mathrm{aut}$. For $\mathrm{BL}\,N_\mathrm{a} < 100\ \mathrm{cm}^{-3}$, the transition time is very low and almost invariant to the other two parameters. As $\mathrm{BL}\,N_\mathrm{a}$
increases, there are joint effects between $\Delta\theta$ and $b_\mathrm{aut}$. For $\mathrm{BL}\,N_\mathrm{a} > 300\ \mathrm{cm}^{-3}$, $\Delta\theta$ and $b_\mathrm{aut}$ have almost linear joint effects
which become mostly invariant to $\mathrm{BL}\,N_\mathrm{a}$. In other words, for very low aerosol concentrations, the aerosol dominates so drizzle
depletion can occur in a wide range of conditions. As aerosol concentrations increase, this effect begins to weaken so low
autoconversion rates can suppress drizzle and strong inversions can reduce decoupling. The lack of transition time dependency
on aerosol concentrations for concentrations above $300\ \mathrm{cm}^{-3}$ could reflect the fact that adding aerosol into already polluted
clouds has a smaller effect than in clean clouds (Carslaw et al., 2013).

### 3.4   Rain water path analysis

We analysed $R$ to determine whether the drivers of the transition might have acted through a drizzle-depletion mechanism. The
PPE $R$ is summarised in Fig. 9, with the domain-averaged timeseries for each member shown in panel a. The PPE is split into
"low" (red) and "high" (blue) $R$ by a temporal mean threshold of $7\ \mathrm{g}\,\mathrm{m}^{-2}$ (approximately half of the highest member). The $f_\mathrm{c}$
for the transitioning simulations (aligned by T0s in panel b and epoch aligned by T1s in panel d) has also been coloured low
and high for $R$ with corresponding subset means. The histograms in panels c and e show the number of points being averaged





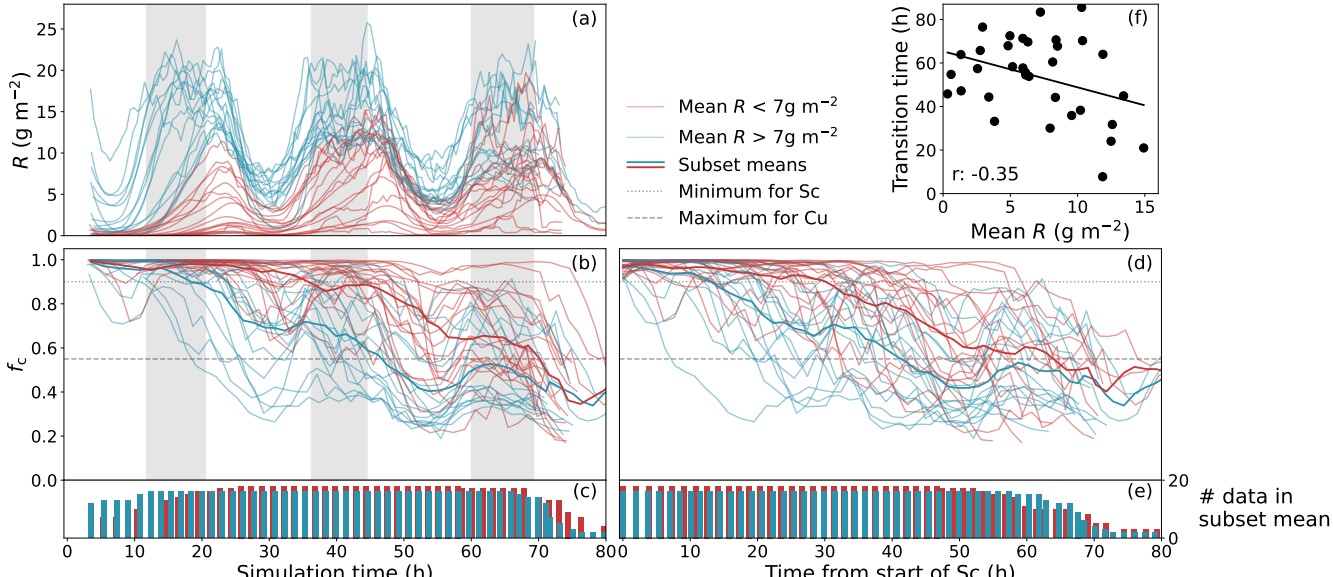

**Figure 9.** The rain water path across the ensemble. a) The domain-averaged rain water path timeseries for each member split by temporal mean rain water path greater than 7 g m$^{-2}$ (blue) or less than (red). b) The cloud fraction timeseries as in Fig. 5c but coloured by mean $R$ The means over each subset (high or low mean R) are shown in bold. c) The number of data points used in calculating the mean of each subset at each timestep in b). d) As in b) but aligned to the start of stratocumulus. e) As in c) but for d). f) A scatter of the mean $R$ for each member against the transition time, with a line of best fit.

over at a given time in each subset, which varies because of the different stratocumulus formation times (Section 3.2 and Fig. 5).

We find that the set of simulations with higher mean $R$ transitioned approximately 22 hours ahead of those with lower mean $R$ (Fig. 9d). Figure 9a shows that those with higher mean $R$ mostly produced drizzle in the first two days, whereas for those with lower mean $R$ the drizzle gradually builds through the simulation. In Fig. 9b, where the timeseries are lined up with the diurnal cycle, the high $R$ subset mean recovers more than the low $R$ mean during the nights. This might suggest that the simulations with more initial rain transition to a state like open-cell stratocumulus rather than cumulus, which would

enable more recovery through the night. In Diamond et al. (2022), they found that drizzle depletion caused the stratocumulus to transition to open-cell behaviour rather than cumulus, but did not determine which factors would cause one transition over the other.

  Figure 9f shows that although the fastest transitions do have a higher mean $R$, drizzle is clearly not the only important factor determining the transition time. Rather, other factors affect the characteristics of the transition, such as the degree of decoupling

and the ability to recover through the night.





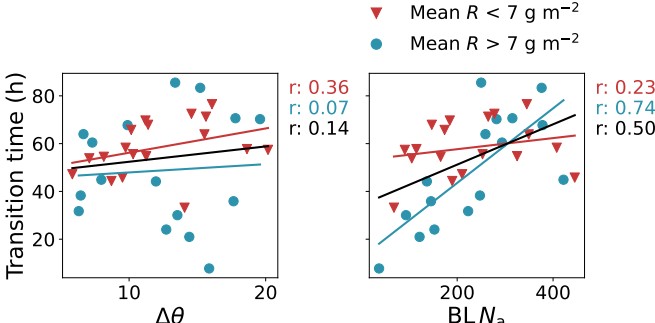

**Figure 10.** One-dimensional scatter plots of $\Delta\theta$ (left) and BL $N_{\mathrm{a}}$ (right) against transition time. The scatter points show the 34 simulations that transitioned within the simulation time and are coloured by high mean $R$ (blue circles) or low mean $R$ (red triangles). Lines of best fit and Pearson's correlation coefficients are calculated for the whole set (black) and each subset.

Figure 10 shows BL $N_{\mathrm{a}}$ has a stronger correlation with transition time when only considering the high mean $R$ cases. Conversely, $\Delta\theta$ has a stronger correlation with transition time when only considering the low mean $R$ cases.

### 3.4.1 Rainwater path average response surfaces

The average response surfaces for the mean $R$ emulator are shown in Fig. 11. The linear contours make it immediately clear

that there are fewer interaction effects compared with the transition time. The mean $R$ has the strongest dependency on BL $z$, with high $R$s in deep layers (panels a, f, g, h, i), which has been found in many previous studies (Bretherton et al., 2010; Eastman and Wood, 2016; Kuan-Ting et al., 2018). The next strongest dependency is on BL $N_{\mathrm{a}}$, with high aerosol producing less rain through precipitation suppression (Albrecht, 1989) (panels d, h, k, m, o). Additionally, there is a strong dependency on $b_{\mathrm{aut}}$ as it is directly linked to the amount of precipitation formed (panels e, i, l, n, o). For both specific humidity parameters,

there is higher $R$ for higher humidity since vapour is available for condensation (BL $q_{\mathrm{v}}$: panels a-e and $\Delta q_{\mathrm{v}}$: panels c, g, j, m, n). Finally, $\Delta\theta$ shows slightly higher mean $R$ under weaker inversions (panels b, f, j, k, l), possibly because weaker inversions are more likely to rise and create deeper boundary layers, which generally drizzle more, but this is a very weak relationship.

The sensitivity analysis of the mean $R$ emulator, shown in top right of Fig. 11, quantifies the effects described above and shows the variance is widely influenced by all parameters rather than being dominated by one specific parameter, like transition

time. The BL $z$ contributes most to the variance in $R$ (35% on average). This is followed by BL $N_{\mathrm{a}}$ (22%), $b_{\mathrm{aut}}$ (19%) and both specific humidity parameters at about 11%. The $\Delta\theta$ contributes less than 1%. The interaction effects are of little importance (2%) in comparison to the three most important parameters. This shows that the mean $R$ is determined more directly by single factors, rather than interactions between them.

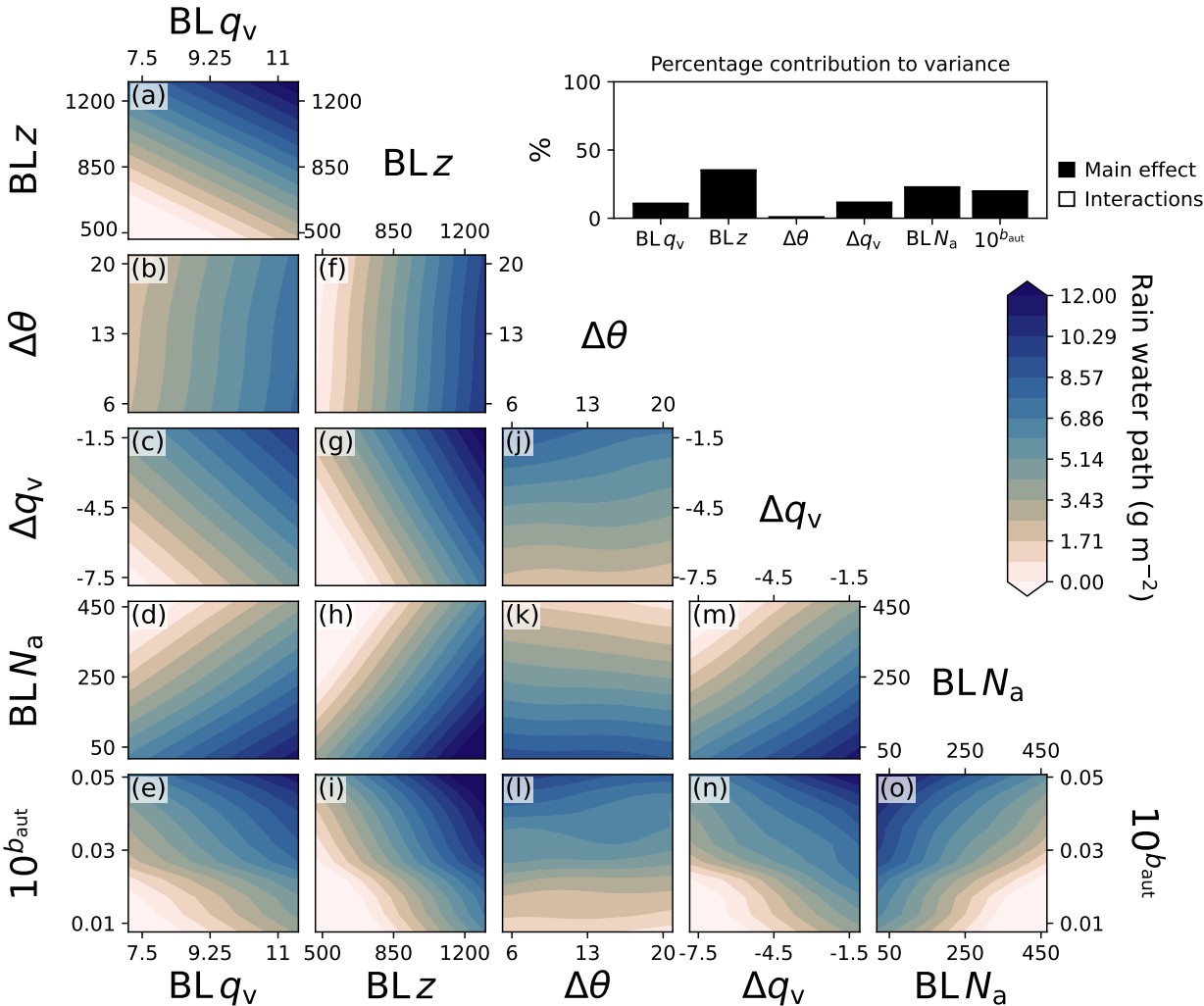

**Figure 11.** Average rain water path response surface sampled with a 1-million point 6-dimensional grid and averaged across hidden dimensions. a-o) shows each 2-dimensional combination of the six perturbed factors averaged in the four dimensions not shown. The inset in the top right shows the contribution of each parameter's variance to the variance in the rain water path.





## 4    Discussion and Conclusions

In this study, we have created a perturbed parameter ensemble of an LES cloud microphysics model with aerosol processing to explore the effects of aerosol and drizzle on the stratocumulus-to-cumulus transition. This novel approach offers a means to investigate the mechanisms underlying the transition and is crucial for assessing the interplay of multiple contributing factors.

We find that aerosol concentration most strongly controls the transition time. In low aerosol environments with less than about $200 \ \mathrm{cm}^{-3}$ the transition time is typically less than 40 hours. These rapid transitions occur in combination with deep

boundary layers or weak inversions, and are more common when using a high autoconversion rate. Boundary layer depth and aerosol concentration most strongly control the mean $R$, followed by the autoconversion rate. Across the full parameter space that we sampled, simulations that have a high mean $R$ transition on average around 22 hours faster than those with a low mean $R$. However, the importance of drizzle varies across the parameter space. The effect of drizzle is particularly strong in the low-aerosol regime, which is consistent with the drizzle-depletion mechanism. However, in the high-aerosol regime drizzle has

a negligible effect and the inversion strength becomes much more important through its determination of entrainment rate and the effect on deepening-warming decoupling.

The PPE approach, with only 34 simulations, effectively captures the joint effects of several cloud-controlling factors in a multi-dimensional parameter space. Where previous studies have focused on the individual effects of parameters, we have identified key combinations of parameters that control the transition time and mean $R$. The PPE approach also reveals that the

part of parameter space with a particularly strong aerosol effect is small, which could explain why fast transitions by drizzle depletion in the real world have not been observed. It is unlikely that campaigns, particularly in the NE Pacific Ocean off the coast of North America, will observe conditions of particularly deep, pristine boundary layers, hence there are no clear observations of a low-aerosol induced rain-hastened mechanism in this region.

The PPE approach exposes other joint effects that were not apparent in previous studies. We find that the inversion strength

has a negligible effect on the transition time in simulations with high mean $R$, whereas in simulations with low mean $R$ it is the second strongest effect (slightly lower than boundary layer moisture, not shown). Previous studies have found that lower tropospheric stability, which is closely linked to inversion strength since it is the difference in potential temperature at 700 hPa and the surface, strongly controls the timing of the transition (Sandu and Stevens, 2011). Our results suggest that this is true when drizzle is playing a minor role in the deepening-warming-decoupling mechanism, but when drizzle depletion is driving

the transition, the inversion strength (and consequently the lower tropospheric stability) has a weaker effect.

Uncertainty in the autoconversion parameter strongly affects the transition time and mean $R$. It is one of the three most important parameters for both. When uncertainty in parameterisations such as this have such a large influence on cloud bulk properties, modelling studies can produce very different results depending on where in parameter space the model lies. An example from Fig. 7 is that low autoconversion rates lower the aerosol concentration at which the transition time becomes

insensitive to aerosol (and so probably more sensitive to inversion strength). The sensitivity of a model to a parameter will be affected by structural differences between models. The effects of structural differences on these sensitivities could be evaluated if other modelling groups were to replicate this work, creating a multi-model PPE.





The details of our results differ from Yamaguchi et al. (2017), but the results support the same conclusions. The drizzle-depletion effect is weaker in our simulations, which is likely due to our model producing less drizzle and also because many of
our simulations form drizzle much earlier, with peaks in the first or second day. This can still cause a drizzle-depletion effect by removing aerosol and moisture from the cloud layer, but it is unlikely to be cumulus-initiated rain causing a positive-depletion feedback because the cumulus generally formed after the second day. The causes of these differences in $R$ are most likely due to differences in domain size or the microphysics scheme. The $R$ values in our simulations are much closer to the values from a sensitivity test in Yamaguchi et al. (2017), which aligns better with our setup, with a domain size of 12 by 12 km$^2$ rather than
24 by 24 km$^2$ and with the Khairoutdinov and Kogan (2000) microphysics scheme rather than the bin-emulating bulk scheme (Fig. 9 and their Fig. 10c). Our study included autoconversion and supports the conclusion of Yamaguchi et al. (2017) that the lack of rain feedbacks in previous studies may partially explain why drizzle was found to have only a minor effect (Sandu and Stevens, 2011; Blossey et al., 2021), and the transition time to be dominated by lower tropospheric stability and entrainment rate.

Compared with other studies that simulated the composite case from Sandu and Stevens (2011), the boundary layer deepening is weaker in our simulations, and this could restrict circulation and precipitation. The maximum height of the boundary layer in our base simulation is around 1400 m, whereas other studies have deepening up to around 2500 m (Sandu and Stevens, 2011; Bretherton and Blossey, 2014; de Roode et al., 2016; Yamaguchi et al., 2017). The previous version of the MONC model was used in the de Roode et al. (2016) model intercomparison, and it has the shallowest boundary layer with a maximum height
of about 1800 m for the reference case (our base case), which suggests that it could be a feature of the MONC model.

Unlike previous studies of the aerosol effect on the stratocumulus-to-cumulus transition, we also included Aitken and coarse mode aerosol. The Aitken buffering hypothesis of McCoy et al. (2021) has been supported by simulations in Wyant et al. (2022) that show Aitken-sized aerosol can be transported to the boundary layer where the larger particles act as cloud condensation nuclei. High concentrations of Aitken mode in the free troposphere slowed the stratocumulus transition to shallow open cells,
which otherwise would have occurred through aerosol removal and precipitation feedbacks. In our simulations, Aitken mode particles are not significantly depleted during the simulations, but this could be a small factor to consider. Additionally, we have not included a source of aerosol through the simulation whereas in reality, sea spray is a primary source of aerosol away from coastal environments. This source would have acted to slow all transitions equally since we did not perturb controlling factors, such as wind speed.

One challenge we faced was how to define a reliable measure of the transition time. This is less of a problem in a small set of simulations that are individually analysed, but it becomes more of an issue when building an emulator that describes the transition time across a multi-dimensional parameter space. As mentioned previously, some of the cumulus clouds may have recovered to stratocumulus after the simulation ended. Similarly for the clouds that began with stratocumulus, there is an unquantifiable amount of time before the simulation where the cloud may have been formed. It may help to spin up a base
cloud before making perturbations and to have a restriction on how long the cloud must remain as cumulus before the end of the simulation. However, perturbations after spinup could cause erratic model responses, and there would still be an adjustment period that would vary across parameter space. Two alternative methods could be to study the time taken for the cloud to



transition from the end point of stratocumulus to the start of cumulus, or the gradients in the decline from stratocumulus. Using $f_c$ is a reliable way to measure a transition in cloud behaviour, but it is difficult to distinguish between an end state of
mesoscale cumulus organisation and open-cell stratocumulus, especially in a domain of this size. Diamond et al. (2022) found open-cell stratocumulus in their study of the transition that used a domain of a similar size, but they did not determine under which conditions the stratocumulus transitioned to a cumulus state or an open-cell state. Despite the small domain size, further analysis of the simulations in this ensemble could give insight into this problem.

The PPE and emulator approach has allowed us to identify joint effects in the stratocumulus-to-cumulus transition, which
create different regimes that align with different mechanisms. The response surfaces also visually showed that the combination of parameters required for the drizzle-depletion mechanism are not typical in the observed regions. In cloud transition studies, being able to understand the occurrence of different regimes under specific parameter combinations is a valuable tool.

*Code and data availability.* All code used to analyse the data and produce the figures in this manuscript may be found on GitHub (Sansom, 2025a). A processed version of the model data is archived on Zenodo and it contains all data used in the analysis (Sansom, 2025b).

*Author contributions.* RS designed the study, ran the simulations and completed the analysis. KS, LL and JJ created the motivation for the study. KS, LL, JJ and LR contributed to discussions and the guided the direction of the analysis. RS prepared the manuscript with input from all co-authors.

*Competing interests.* At least one of the (co-)authors is a member of the editorial board of Atmospheric Chemistry and Physics.

*Acknowledgements.* This work was possible only thanks to the ARCHER2 UK National Supercomputing Service (Beckett et al., 2024,
https://www.archer2.ac.uk), which was used to run all simulations. The analysis and storage of data was all completed using JASMIN, the UK's collaborative data analysis environment (Lawrence et al., 2013, https://www.jasmin.ac.uk). RS and KS received funding from the EPSRC DTP (grant no. 2114653). RS, KC, LL and LR were funded by the European Union's Horizon 2020 research programme (FORCeS (grant no. 821205)). LR, JJ and KC were funded by NERC Aerosol-MFR (grant no. NE/X013901/1). LR was supported by the Met Office Hadley Centre Climate Programme funded by DSIT. RS is grateful for the use of the Met Office/NERC cloud model and the assistance from
Adrian Hill, Adrian Lock at the Met Office and Steef Böing, at the University of Leeds. The figures in this manuscript were produced using colour maps from Scientific colour maps, developed to tackle the misuse of colour in scientific communication and making sure figures are readable by all (Crameri et al., 2020; Crameri, 2023).



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
