# Peer review of "Strong control of the stratocumulus-to-cumulus transition time by aerosol: analysis of the joint roles of several cloud-controlling factors using Gaussian process emulation"

_EGUsphere, 2025_

## Author Response (AR1)

**Response to Reviewers**

Manuscript title: Strong control of the stratocumulus-to-cumulus transition time by aerosol: analysis of the joint roles of several cloud-controlling factors using Gaussian process emulation

Corresponding author: Rachel W. N. Sansom

We thank both reviewers for the generous amount of time and effort they put into reviewing this manuscript. We are pleased that they think the study is valuable and we have strived to include their suggestions to improve the manuscript. In particular, we think the addition of a sensitivity test on the cumulus threshold and addition aerosol analysis really strengthens the study. Changes to text are shown as in "tracked changes", where underlined segments are new text and scored out text has been deleted. Line numbers refer to the revised version of the manuscript.

**Reviewer 1:**

*This study investigates the role of aerosol concentration and other cloud-controlling factors in the stratocumulus-to-cumulus transition using a perturbed parameter ensemble (PPE) of large-eddy simulations and Gaussian process emulation. By sampling a wide range of environmental conditions, the authors systematically evaluate both individual and joint effects of key parameters, such as boundary-layer aerosol, inversion strength, and autoconversion rate, on transition time and rain water path. This approach allows for a more comprehensive understanding of the processes driving the transitions and the conditions under which mechanisms like drizzle depletion become dominant.*

*In my opinion, the LES and emulation framework are both comprehensive and valuable. The manuscript is well-structured, and the analyses are presented clearly. That said, several clarifications and expanded discussions would further improve the manuscript, and I have outlined these in the comments below. I recommend publication after these points are addressed.*

**General Comments:**

**The exclusion of SST as a perturbed parameter may limit the exploration of the deepening–warming transition pathway relative to the drizzle-depletion mechanism. Although the consequence of excluding SST is mentioned briefly in the Results section, its potential importance deserves further discussion in the Conclusions to highlight how its omission may affect the balance of transition pathways represented.**

It is well established that increasing SST is the main driving factor behind both transition mechanisms along these trajectories. We have added extra text to highlight that SST is the main driving factor. The motivation for this research was to analyse the various contributions of other key factors compared with aerosol concentration. Including SST as a parameter would have expanded the paper to cover cloud feedback, which is beyond what we were trying to understand here. Additionally, we focused on trying to identify conditions under which the drizzle-depletion mechanism might occur, so parameters related to drizzle were

prioritised. We agree that this does affect the balance of transition pathways represented and have made some adjustments to the text to reflect this point.

**Line 1:** Stratocumulus-to-cumulus transitions are driven primarily by increasing sea-surface temperatures, with additional contributions from numerous interacting cloud-controlling factors.

**Line 4:** We consider the role of aerosol together with several other cloud-controlling factors representing a selection of the wider environmental conditions that affect drizzle.

**Line 11:** We find that in the low-aerosol regime (< 200 cm^{-3}) the transition time is most strongly affected by the aerosol concentration out of the factors considered here.

**Line 168:** Although SST varies along the airmass trajectory, we chose not to include perturbations to SSTs or SST gradients among the parameters we investigated. To be useful, such a study focusing on cloud feedback would need to consider realistic covariations of SSTs with the cloud controlling factors under investigation.

**Line 452:** We find that aerosol concentration most strongly controls the transition time out of the factors considered here.

**Line 461:** The PPE approach, with only 34 simulations, effectively captures the joint effects of several cloud-controlling factors in a multi-dimensional parameter space. Where previous studies have focused on the individual effects of parameters, we have identified key combinations of parameters that control the transition time and mean R. The PPE approach also reveals that the part of parameter space with a particularly strong aerosol effect is small, which could explain why fast transitions by drizzle depletion in the real world have not been observed. It is unlikely that campaigns, particularly in the NE Pacific Ocean off the coast of North America, will observe conditions of particularly deep, pristine boundary layers, hence there are no clear observations of a low-aerosol induced rain-hastened mechanism in this region. However, "ultra-clean layers" where the concentration of particles larger than 0.1 $\mu$ m is below 10 cm−3, are a common feature of the transition and may be the result of the drizzle-depletion mechanism (Wood et al., 2018; Kuan-Ting et al., 2018). We have also only considered 6 dimensions out of a much larger multi-dimensional problem. With the inclusion of other variables that could have a larger influence on the deepening-warming mechanism (such as initial SST, subsidence or wind speeds) the region with a strong aerosol effect is likely smaller than what we have shown here.

***The calculation of output variable, transition time, is subject to noise, and while this is acknowledged in the manuscript, additional clarifications would be helpful. Specifically, the definition of transition time based on cloud fraction (fc) thresholds of 0.9 and 0.55 appears somewhat subjective. It would really strengthen the study to perform a sensitivity analysis using alternative fc thresholds and report the resulting differences in transition time.***

We think the threshold of 0.9 is fair for stratocumulus cloud, but we acknowledge that the cumulus threshold of 0.55 may appear arbitrary. Since we have a finite simulation time and a lower threshold value intrinsically means a later transition, using a lower threshold means fewer simulations qualify before the simulation end is reached. The value 0.55 was initially

chosen as a reasonable value for a cloud transition (from 0.9) that maximised the number of simulations that qualified. We have now done some sensitivity tests to show that the key relationships we discuss in the text still hold for a lower cumulus threshold. We show this using the adapted version of Fig. 10, where Fig. 10a has a cumulus threshold of 0.55, as before, and Fig. 10b has a cumulus threshold of 0.47. The correlation of transition time with delta theta in low mean R and the correlation of transition time with BL N_a for high mean R are both significant even with this lower threshold. Beyond this threshold, the correlations are still apparent but lose statistical significance because of a reduction on the size of the qualifying dataset.

[Figure]

**Figure 10a:** One-dimensional scatter plots of $\Delta\theta$, BL Na and R against transition time for a cumulus cloud fraction threshold of 0.55. The scatter points show the 34 simulations that transitioned within the simulation time and are coloured by high mean R (blue circles) or low mean R (red triangles). Lines of best fit, Pearson's correlation coefficients (r) and statistical significance (p) are calculated for the whole set (black) and each subset.

[Figure]

**Figure 10b**: One-dimensional scatter plots of $\Delta\theta$, BL Na and R against transition time for a cumulus cloud fraction threshold of 0.47. The scatter points show the 28 simulations that transitioned within the simulation time and are coloured by high mean R (blue circles) or low mean R (red triangles). Lines of best fit, Pearson's correlation coefficients (r) and statistical significance (p) are calculated for the whole set (black) and each subset.

We have added Fig. 10b in the appendix with the following text:

Figure A1 shows a repetition of the 1-dimensional parameter analysis from Fig. 10 to determine whether the key correlations still hold for a lower cumulus threshold. Here, the threshold for cumulus cloud has been reduced to a f_c of 0.47. Reducing this threshold results in a mean ensemble transition time of 57 hours, which is 3 hours longer than for a

cumulus threshold of 0.55. The significant correlations in Fig. 10 are still significant with the lower threshold. The correlation of transition time with delta theta is slightly stronger and the correlation of transition time with BL N_a is slightly weaker.

**Line 206:** Figure 2 shows two examples of how this was calculated from the cloud fraction (fc) for all the ensemble members based on fc > 0.9 for stratocumulus and fc < 0.55 for cumulus. The value of 0.55 for cumulus was chosen as a reasonable value for a cloud transition that maximised the number of transitioning ensemble members available for emulation. The sensitivity test in Appendix A shows that the key conclusions are statistically significant down to a threshold f_c of 0.47, after which not enough simulations transition within the simulation to be significant.

***The model configuration, including the reference trajectory, forcings, and boundary conditions, is idealized, and this limits the realism of the simulations. Although the manuscript notes this briefly in the "perturbation method" section, it should be stated more explicitly in Section 2.1 and reiterated in the Conclusions.***

We agree that this is an important point about the study. The following sentences have been added to highlight this aspect.

In the abstract, **line 7:**

A 34-member perturbed parameter ensemble of idealised large-eddy simulations with 2-moment cloud microphysics is used to train Gaussian process emulators…

At the end of the LES setup description on **line 154:**

The LES setup is idealised because realistic profiles would be specific to an individual transition case rather than being generally representative of a typical case. Although this may limit the realistic nature of the simulations, it simplifies the perturbation method for a study such as this where perturbations are made from a reference case to learn broadly about the transition behaviour across parameter space. This idealised setup also enabled comparison with previous studies that used the same approach. (e.g. Yamaguchi et al., 2017).

***All results presented are based on idealized modeling, which makes it difficult for direct comparison with observations. A more detailed discussion would be helpful on how the results can be interpreted considering this limitation. For example, to what extent can we trust the results of this study? Could some be artifacts of the model structure? Have previous studies assessed the fidelity of this LES model and its microphysics scheme for stratocumulus-to-cumulus transitions?***

These are very valid questions about idealised modelling. We have based this research on the large set of literature surrounding the shallow cloud studies using LES models, which was an important reference point when extending the research to complex PPEs. Although individual trajectory cases are often compared to specific observations, it is also quite common for an idealised set up to be used as with all the studies that simulate the Sandu and Stevens (2011) composite case. With these idealised simulations, the idea is that key features of the cloud transition are captured without dwelling too much on the nuances of a

particular case. For a study such as this, where we are considering the general behaviour throughout parameter space using an ensemble, we think idealised modelling is an appropriate method.

We acknowledge that without comparison to an observational case it is then difficult to know the extent of the model's influence on the results. We have dedicated section 3.1 to comparing the results of the base simulation with previous studies that have simulated the same composite case, one of which includes results from a former version of this model that took part in an intercomparison project. We have discussed the main differences in the lower rain water path and the shallower boundary layer and included mentions of these points in the discussion of the results. With the help of both reviewers, we have made quite a few additional clarifications and caveats, which we feel has really strengthened the manuscript.

***I am not sure if generalizing the results would be straightforward, especially since they are based on a specific region and season. It would improve the manuscript to clearly state the geographical location (Subtropical Northeastern Pacific) and season (currently not mentioned in the manuscript) both in the Abstract and the Conclusions. While the exclusion of semi-direct aerosol effects and highly polluted conditions is noted briefly, this should be clarified in the Conclusions.***

Thank you for pointing out that this important information was missing from the manuscript. It has now been added in the description of the simulation (see later comment) and in the abstract and conclusions.

**Line 9**: … and average rain water path. We base the ensemble around a composite of trajectories in the Northeastern Pacific during summer. Using these emulators…

**Line 447:** … on the stratocumulus-to-cumulus transition. The ensemble is based on the Sandu and Stevens (2011) composite case, which was created to represent a typical trajectory in the NE Pacific during summer. This novel approach offers a means to investigate the mechanisms underlying the transition and is crucial for assessing the interplay of multiple contributing factors. It should be noted that include highly polluted aerosol conditions and the effect of the semi-direct aerosols in plumes is beyond the scope of this study.

**Specific Comments:**

***L21–23**: It would make sense to change this sentence to something like: "Low clouds in the subtropics have a cooling effect on the planet. However, global climate models (GCMs) project a future decrease in their cloud fraction, which would reduce that cooling effect, amplify warming, and contribute to a positive cloud feedback."*

Thank you for this suggestion, the paragraph has been altered.

**Line 21**: Uncertain processes lead to poor parameterisations in global climate models (GCMs) so transitions are not captured well, which creates large uncertainties in simulated cloud properties and their responses to the warming climate (Bony and Dufresne, 2005; Teixeira et al., 2011; Eastman et al., 2021). Low clouds in the subtropics have a cooling effect on the planet, and since GCMs project a future decrease in subtropical cloud fraction,

that cooling effect will be weakened amplifying warming, and contributing to a positive cloud feedback effect (Bretherton, 2015; Ceppi et al., 2017; Nuijens and Siebesma, 2019).

**L69-70:** *Please check the grammar.*

Thank you for highlighting this.

**Line 72:** Eastman et al. (2022) assessed the difference between closed-cell stratocumulus that do and do not transition.

**L79**: *Is there a better term to use here in place of "calculated"? I am not sure to understand it here.*

In this sentence we are referring to simulated Lagrangian trajectories, which are often referred to as being calculated or computed. "Computed" is perhaps more fitting so we have changed to that.

**Line 80:** Small perturbations to initial conditions can represent different stages of the transition (Chung et al., 2012; Tsai and Wu, 2016; Bellon and Geoffroy, 2016), while simulating observed or computed trajectories with completely different sets of initial conditions produces very different transition characteristics (Goren et al., 2019; Blossey et al., 2021; Erfani et al., 2022).

**Line 111:** Sandu et al. (2010) computed thousands of forward and backward air parcel trajectories…

**L84-92**: *Please provide more detailed descriptions of Gaussian process emulation and PPE, how they are related, and how they benefit the study of clouds and their transitions, so the concepts are more understandable to readers unfamiliar with these methods.*

We have rewritten this paragraph to be more accessible.

**Line 86:** Using machine learning, "emulators" can statistically represent the multi-dimensional relationship between a set of cloud-controlling factors (parameters) and a specific cloud property. The behaviour of complex cloud models can be efficiently sampled to create training data using a perturbed parameter ensemble (PPE) approach, where parameters are perturbed in combination, rather than one at a time. This method provides sufficient information with a sparse sampling of the multi-dimensional parameter space, which is ideal for emulating computationally expensive models. Gaussian process emulation works well with relatively few points compared to other machine learning methods (10s or 100s as opposed to 1000s) (O'Hagan, 2006). Once validated, the emulators, can be used to fill the multi-dimensional parameter space with predictions. This dense sampling can then be used for sensitivity analysis to quantify the contributions from each factor to the variance in the property (Saltelli et al., 2000; Johnson et al., 2015; Wellmann et al., 2018, 2020) or to create response surfaces, which enable us to visualize non-linear joint effects of factors or the relationships between cloud states, e.g., Glassmeier et al. (2019) and Hoffmann et al. (2020). The PPE method with emulation is well suited to identifying distinct behaviour regimes in cloud models (e.g., Johnson et al., 2015; Sansom et al., 2024).

*L92: Change "in Sansom et al. (2024) we used" to "Sansom et al. (2024) used."*

Agreed and changed.

*L106: Mention the period of the study (date, year, etc.) in Sandu et al. (2010), which your study is based on.*

Thank you for this suggestion. Details of the study in Sandu et al. (2010) have now been added.

**Line 111**: The PPE is based on the composite case created for the NE Pacific Ocean basin in Sandu and Stevens (2011). Sandu et al. (2010) computed thousands of forward and backward air parcel trajectories from areas of extensive cloud cover between May and October for 2002 to 2007. Boundary layer properties were retrieved over a six day period of advection from satellite data and meteorological reanalysis.

*L108: So, is the reference case an average among many trajectories?*

Yes indeed, we have expanded on the description of the composite case to make this clearer in the text.

**Line 114:** Sandu et al. (2010) found that the climatological, or averaged, trajectory represented the key characteristics of the transition well. Sandu and Stevens (2011) developed this into a reference case for numerical simulation that represents a typical trajectory in the NE Pacific Ocean for June to August in 2006 and 2007 from a subset of trajectories for the three days in which the majority of the transition occurred (Sandu and Stevens, 2011).

*L112–118: Mention the MONC and CASIM versions, if possible.*

Agreed and added.

**Line 120:** The ensemble was simulated using the UK Met Office and National Environmental Research Council (NERC) LES model, called the MONC (Met Office/NERC Cloud) model (Dearden et al., 2018; Poku et al., 2021; Böing et al., 2019). The model solves a set of Boussinesq-type equations, using an anelastic approximation here, which is based on a reference potential temperature profile that depends only on height. The subgrid turbulence parameterization is an extension of the Smagorinsky-Lilly model and is based on that described in Brown et al. (1994). Version 0.9.0 of the Leeds-MONC Github repository was adapted for this study and released as version 0.9.1 (Denby et al., 2025). Here, MONC was coupled to the two-moment Cloud AeroSol Interaction Microphysics scheme (CASIM, version 6341: Shipway and Hill, 2012; Hill et al., 2015) and the Suite of Radiation Transfer Codes based on Edwards and Slingo (SOCRATES, version 1012: Edwards and Slingo, 1996).

*L121: You mentioned wind profiles are retained. How about temperature and humidity (assuming they are used as forcing) in the boundary layer and free troposphere? Also, is subsidence a forcing in your LES, or is it constant for all runs?*

We have added this information.

**Line 143:** Wind profiles were retained to ensure appropriate ocean surface evaporation, but the model has periodic boundary conditions so the domain was always focused on the same cloud cell. The temperature and specific humidity profiles were allowed to evolve freely and the large-scale divergence was set to a constant value of $1.86 \times 10^{-26}$ s$^{-1}$. The large-scale subsidence is calculated in the model as -Divergence x vertical height above sea level.

*L123: What is the spin-up period in your simulations? Also, specify the initial and final SST values in your study.*

We have added this information.

**Line 147:** Simulations were run for 3-4 days with a spin-up period of around an hour being discarded. The SST was increased by nearly 1.5 K per day, from 293.75 K to 300.93 K, following Sandu and Stevens (2011), Bretherton and Blossey (2014) and Yamaguchi et al. (2017).

*L127-138: It would make more sense if this paragraph were moved up right after introducing CASIM. Also, I assume CASIM is only active in the boundary layer, correct? Then, how does it interact with aerosols in the free troposphere? Did you use a constant or time-varying value for the free-tropospheric aerosols? Does CASIM have a surface source of aerosol in your study? These can be clarified in the manuscript.*

We agree that this paragraph is better suited after the introduction of CASIM so we have moved it to line 119.

Aerosols in the free troposphere can be entrained into the cloud and then processed by CASIM. We used a constant free-tropospheric aerosol, which is mentioned on line 166, but we have added the concentrations and clarified that there is no surface source of aerosol in this study, unlike in Yamaguchi et al. (2017).

**Line 195:** The initial boundary layer concentration of accumulation mode aerosol was perturbed because the vast majority of aerosols that activate into cloud droplets (cloud-condensation nuclei) are from the accumulation mode. Boundary layer Aitken mode was initialised with a concentration of 150 cm$^{-3}$ and allowed to freely evolve. Free-tropospheric aerosol can also be a source of cloud-condensation nuclei and could be important in simulations with very low aerosol concentrations in the boundary layer (Wyant et al., 2022). However, free-tropospheric aerosol concentration was kept constant across the PPE because it was not expected to be as important as the key factors chosen. The Aitken concentration was 200 cm$^{-3}$ and the accumulation concentration was 100 cm$^{-3}$. There is no surface source of aerosol throughout the simulations.

*L130: Can you add the particle size distribution (normal, gamma, etc.) for the Aitken, accumulation, and coarse modes?*

Thank you for pointing this out, we have added it accordingly.

**Line 133:** The Aitken mode distribution has a standard deviation of 1.25 and a mean radius of 25 nm. The accumulation mode distribution has a standard deviation of 1.5 and a mean radius of 100 nm. All aerosol size modes are represented by a lognormal distribution.

***Section 2.2 and Table 1****: Some explanation is needed regarding how these six variables and their ranges were selected. Were they based on previous studies? Have you done sensitivity tests to exclude other variables? As I mentioned earlier, SST is a very important variable especially for deepening-warming transition. Surface wind speed and subsidence at the inversion level are also critical parameters that excluded. Also, you should clarify whether the values of these variables are chosen at the initial time or averaged along the trajectories.*

The five cloud-controlling environmental parameters were chosen based on sensitivity tests from previous studies that have been cited in the paragraph beginning at line 192. Because this is the first PPE of several cloud-controlling factors, we used the same factors (parameters) as previous studies to maintain consistency. We also wanted to include an uncertain model parameter for the autoconversion since we were interested in the drizzle-depletion mechanism. SST was not included for the reasons given above. The values of the parameters were initially perturbed based on the initial value, but because some clouds did not develop until later in the study, the final values used were the values at the beginning of stratocumulus cloud (T1).

The paragraph that was beginning at line 192 ("The parameter ranges were chosen…") in section 2.2.1 has been moved to section 2.2. We have also added a couple of sentences in the discussion about the balance of pathways represented in this parameter space.

**Line 159:**

**2.2 Perturbed parameter ensemble**

PPEs are a valuable tool for understanding the joint effects of parameters on model output. Perturbing parameters simultaneously in a space-filling way maximizes information from the model about how parameters jointly affect the outputs of interest. Five cloud-controlling factors were perturbed, plus a sixth factor that alters the dependence of the autoconversion rate on $N_d$. Table 1 shows the individual ranges for each parameter, which form the boundaries of the 6-dimensional hypercube that the ensemble covers.

The parameter ranges were chosen to span the breadth of studies on stratocumulus-to-cumulus transitions in the subtropics. Often case studies are designed for LES simulation from observations of particularly fast or slow transitions, so a broad range of behaviours was included in the parameter space by spanning these reported cases (Sandu and Stevens, 2011; de Roode et al., 2016; Blossey et al., 2021). Although SST varies along the airmass trajectory, we chose not to include perturbations to SSTs or SST gradients among the parameters we investigated. To be useful, such a study focusing on cloud feedback would need consider realistic covariations of SSTs with the cloud controlling factors under investigation. Since many LES studies have not focused on the aerosol effect, the range for the accumulation mode concentrations was informed by the Cloud System Evolution in the Trades (CSET) and Marine ARM GPCI Investigation of Clouds (MAGIC) campaigns (Bretherton et al., 2019; Painemal et al., 2015). Note that we have not included extremely polluted cases, such as the biomass burning region off the western coast of Africa. There are many studies of the aerosol semi-direct effect on the stratocumulus-to-cumulus transition in the Atlantic ocean, with some contradicting results (Yamaguchi et al., 2015; Zhou et al.,

2017; Diamond et al., 2022). Further understanding of transition mechanisms will help to untangle these joint effects.

**Line 461:** The PPE approach, with only 34 simulations, effectively captures the joint effects of several cloud-controlling factors in a multi-dimensional parameter space. Where previous studies have focused on the individual effects of parameters, we have identified key combinations of parameters that control the transition time and mean R. The PPE approach also reveals that the part of parameter space with a particularly strong aerosol effect is small, which could explain why fast transitions by drizzle depletion in the real world have not been observed. It is unlikely that campaigns, particularly in the NE Pacific Ocean off the coast of North America, will observe conditions of particularly deep, pristine boundary layers, hence there are no clear observations of a low-aerosol induced rain-hastened mechanism in this region. We have also only considered 6 dimensions out of a much larger multi-dimensional problem. With the inclusion of other variables that could have a larger influence on the deepening-warming mechanism (such as initial SST, subsidence or wind speeds) the region with a strong aerosol effect is likely smaller than what we have shown here.

*L177: "Latin Hypercube" is a critical component of your study, so it would be helpful to dedicate a paragraph or a few sentences briefly describing it and justifying its use. What is its benefit compared to assigning equally spaced values within the range of each variable?*

The Latin hypercube approach more efficiently samples the model behaviour than a grid. In a grid approach, multiple simulations use the same values for parameters instead of learning about how that parameter affects the output in a new part of parameter space in every simulation. We have added a couple of sentences justifying its use. We have talked about it being a space-filling design and it is a well-documented method, so we leave the reader to discover the details of the method in the references.

**Line 211:** The perturbation values were chosen using a "maximin" Latin hypercube approach. Figure 1 shows the 6-dimensional design, which maximizes the minimum distance between points to ensure that values are well-spaced across the multi-dimensional parameter space and unique along each parameter axis (Morris and Mitchell, 1995; Jones and Johnson, 2009). Perturbing parameters simultaneously whilst ensuring uniqueness in every dimension ensures that each simulation provides valuable new information about the model behaviour across parameter space, especially if some dimensions (parameters) do not affect the model output. Crucially, this allows sufficient sampling of parameter space with a smaller number of simulations than a grid approach.

*L182: Here, you correctly mention that the model setup is idealized. This should also be stated when describing the LES setup and reiterated in the Conclusion.*

The following sentence has been added at the end of the LES setup description:

**Line 149:** The horizontal resolution was 50 m, and the vertical resolution varied from 20 m near the surface, to 5 m around the temperature inversion, and gradually increased above that. The LES setup is idealised because realistic profiles would be specific to an individual transition case rather than being generally representative of a typical case. Although this may limit the realistic nature of the simulations, it simplifies the perturbation method for a

study such as this where perturbations are made from a well-studied reference case to learn broadly about the transition behaviour across parameter space.

The first sentence of the Conclusion has been altered to reiterate the idealised nature of the simulations:

**Line 446:** In this study, we have used an LES cloud microphysics model with aerosol processing to create an idealised perturbed parameter ensemble and explore the effects of aerosol and drizzle on the stratocumulus-to-cumulus transition.

*L189–191*: It is important to explain this "new understanding." What are the ranges of each variable that result in stratocumulus clouds and a transition within your setup?

Initially Fig. 1 showed how the parameter values shifted after spin up, but it has now been altered to show the parameter values at the beginning of stratocumulus formation. This makes it clearer that there were combinations of Bl qv and Bl z that did not allow stratocumulus to form. Table 1 has been updated to show the range values at this time also. The phrase "new understanding" has been changed to be less ambiguous:

**Line 227:** These points were augmented  to fill the regions of parameter space that produced stratocumulus and were likely to transition within simulation time, increasing the density of information in the most relevant part of parameter space.

**Line 220:** The perturbed cloud-controlling factors evolved during model spinup and, in some simulations, before a stratocumulus cloud formed. Although the parameter space changed, the points remained spaced well enough for emulating, so we analysed the relationships between the values at the beginning of stratocumulus and the transition properties.

**Updated Figure 1:**

[Figure]

**Figure 1:** The Latin hypercube design for the 34-member perturbed parameter ensemble. Each 2-dimensional plot shows a different combination of two of the six parameters over the chosen ranges (see Table 1). The grey circles show the values used for the initial conditions in each simulation from the original 97-member Latin hypercube design and the black points show how these values shifted in the 34 members that developed stratocumulus and transitioned. The inset shows how the parameters are perturbed in the initial profiles using this design.

**Updated Table 1:**

Table 1. Parameter descriptions, symbols, designed range in parameter space and shifted range at the beginning of stratocumulus formation.

| Parameter description | Symbol | Designed range | Range at Sc |
|---|---|---|---|
| Boundary layer vapor mass mixing ratio | BL $q_v$ | 7 to 11 g kg$^{-1}$ | 8.0 to 12.0 g kg$^{-1}$ |
| Boundary layer depth | BL $z$ | 500 to 1300 m | 467.9 to 1280.8 m |
| Inversion jump in potential temperature | $\Delta\theta$ | 2 to 21 K | 4.9 to 20.1 K |
| Inversion jump in vapor mass mixing ratio | $\Delta q_v$ | -7 to -1 g kg$^{-1}$ | -8.6 to -1.8 g kg$^{-1}$ |
| Boundary layer aerosol concentration | BL $N_a$ | 10 to 500 cm$^{-3}$ | 33.5 to 447.4 cm$^{-3}$ |
| Autoconversion rate parameter (Khairoutdinov and Kogan, 2000) | $b_{aut}$ | -2.3 to -1.3 | -2.1 to -1.3 |

*L191: Add the final number of total experiments and the number of transitioning experiments.*

Agreed and added.

**Line 230:** In total 97 simulations were run with a final 34 simulations showing cloud transitions that matched our definition of a stratocumulus-to-cumulus transition.

*L197: Mention the locations of these campaigns (Subtropical Eastern Pacific?).*

Agreed and added.

**Line 170:** Since many LES studies have not focused on the aerosol effect, the range for the accumulation mode concentrations was informed by the Cloud System Evolution in the Trades (CSET) and Marine ARM GPCI Investigation of Clouds (MAGIC) campaigns, which took place in the NE Pacific (Bretherton et al., 2019; Painemal et al., 2015).

*L198: This is an important point and should be mentioned in the Abstract and/or Conclusions: this study considers clean to moderately polluted aerosol conditions.*

A few words have been added to the abstract to clarify this point:

**Abstract:** Stratocumulus-to-cumulus transitions are driven primarily by increasing sea-surface temperatures, with additional contributions from numerous interacting cloud-controlling factors. Understanding these interactions is important for improving the accuracy of cloud responses to changes in climate and other environmental factors in global climate models. Many studies have found lower-tropospheric stability dictates the transition time, while aerosol-focused studies found that aerosol concentration plays a key role via the drizzle-depletion mechanism. We consider the role of aerosol together with several other cloud-controlling factors representing a selection of the wider environmental conditions that affect drizzle in a clean to moderately polluted environment. A 34-member perturbed parameter ensemble of idealised large-eddy simulations with 2-moment cloud microphysics is used to train Gaussian process emulators (statistical representations) of the relationships between the factors and two properties of the transition: transition temporal length and average rain water path. We base the ensemble around a composite of trajectories in the Northeastern Pacific during summer. Using these emulators, parameter space can be densely sampled to visualise the joint and individual effects of the factors on the transition properties. We find that in the low-aerosol regime ($< 200$ cm$^{-3}$) the transition time is most strongly affected by the aerosol concentration out of the factors considered here. Fast transitions, under 40 hours, occur in this regime with high mean rain water path, which is consistent with a drizzle-depletion effect. In the high-aerosol regime, the inversion strength becomes more important than the aerosol concentration through the inversion's effect on entrainment and the deepening-warming decoupling mechanism.

*Figure 1 caption: Mention the total number of experiments shown here.*

See updated Fig. 1 above.

*L205–206: Are these thresholds rather arbitrary? Have you done sensitivity tests to define transition time based on different values?*

As discussed above, we have now done some sensitivity tests to show that the key correlations hold for a lower cumulus threshold value. Statistical significance is lost below $f\_c$ = 0.47 because not enough simulations qualify before the end of the simulation.

*Figure 3: Add the total number of emulators and also the number of MONC simulations. Also, clarify that rain water path is averaged over each simulation (if I understood correctly from the text).*

This validation is of the two emulators shown in Fig. 3, one for transition time and one for the mean R. Repetition of the emulation code produces consistent emulator results. The caption has been changed to include information on the total number of simulations used and the averaging of the rain water path. The rain water path title now reads "Mean rain water path" and the panels have labels.

[Figure]

**Figure 3:** Emulator validation using the leave-one-out approach. a) Transition time and b) rain water path averaged over the transition. Both emulators were trained with the 34-member PPE. Points show the model output against the emulator-predicted values for each training data point that has been left out of the emulator training set in turn. Lines show the upper and lower 95% confidence bounds. Black points are where the model output data lies within the confidence bounds (pass) and red points are where this is not the case (fail).

*L254–255: To be more accurate, change this to: "shows three snapshots of liquid water path (L) and water mass mixing ratio (MMR) from the beginning, middle, and end of the simulation, along with time series of fc, L, and rain water path (R)."*

The description of this figure in the main text has been updated along with the figure and this is no longer an issue.

*L267: Do any of those studies provide observations for comparison?*

The simulated case is a composite case so there are no observations for comparison. We have instead compared against their simulations of the base case to see where ours differs.

*L270: Remind the reader that Yamaguchi et al. (2017) uses an aerosol-aware cloud microphysics scheme similar to yours. It would be helpful to comment on which of the other studies mentioned above also use an aerosol-aware scheme. Also, are Yamaguchi et al. (2017) sensitivity tests based on bulk microphysics?*

We have clarified the information about the microphysics schemes.

**Line 306**: This could be due to the different radiation schemes and mixing processes in the models, or to the stretching of the vertical layers in the top of the domain. Yamaguchi et al. (2017) is the only study using aerosol processing that we compared our base simulation to. In our simulation, R peaks at about 25 g m$^{-2}$ at the beginning of the third day, which aligns roughly with the sensitivity test in Yamaguchi et al. (2017) that used a similar domain size to ours with the bulk microphysics scheme from Khairoutdinov and Kogan (2000). However, it is much less than the peak of 150 g m$^{-2}$ for the same domain size using their bin-emulating bulk microphysics scheme.

*Figure 4: Since you discuss boundary layer depth in the text, can you show it on the vertical cross-sections? You defined fc as MMR greater than 0.01. Can you confirm this is the lowest MMR value shown in the cross-sections? Since this case is well studied, I assume observational (at the very least, satellite) exploration has been done by others. It would make sense to compare your simulations with those observations and discuss the fidelity of your model in simulating this case.*

Figure 4 now includes the inversion height and time series of aerosol concentrations in the boundary layer. The MMR is indeed masked out at a threshold of 0.01 g kg$^{-1}$ and this is now included in the colourbar and the caption. The colourmap for L is now logarithmic to distinguish better between areas of low L and typical L. Labels are also now included for each panel:

[Figure]

**Figure 4.** Base simulation cloud properties.  a-d) timeseries of cloud fraction (f_c), liquid water path (L), rain water path (R), and boundary layer aerosol concentrations (N_a). Grey shading indicates local nighttime. e-j) Snapshots at 9pm local time on day 1 (e-f), day 2 (g-h) and day 3 (i-j). Top row (e, g, i) shows top-down views of L and bottom row (f, h, j) shows a vertical cross section of liquid water mass-mixing ratio (MMR) at the y-location of the transect line. The MMR is masked for values lower than 0.01 g kg$^{-1}$, in line with the f_c definition.

As mentioned above, the studied case is a composite case so there are no direct observational data. As discussed, our simulations are idealised and not meant to recreate a single real transition with high fidelity. Rather, they are meant to capture key elements of the

transition under different environmental conditions to understand the relative contributions of different factors. This is a key concept of our study and rooted in many of the LES studies mentioned in this manuscript.

***Figure 4 caption****: Add "path" after "Snapshots of liquid water."*

See above comment for updated Fig. 4 caption.

***L278–279****: When you mention "subset mean in Fig. 5c is a similar shape to the base simulation," please add that the transition in the base simulation occurs later compared to the mean. Also, change "is" to "has" in that sentence.*

Agreed, this suggestion has been included. Figure 5 is now figure 6 due to the addition of a figure summarising the PPE (see later comment from reviewer 2).

**Line 331**: The subset mean in Fig. 6c has a similar shape to the base simulation, but the subset mean transitions a few hours earlier. However, the PPE members show a wide range of behaviours.

***L286–291****: You should add that the number of simulations with warm SST is not enough to draw a definitive conclusion. Also, why not select SST as a seventh parameter? SST is such an important factor in the transition that leaving it out needs justification. This relates to the earlier comment on the criteria for including or excluding parameters.*

Agreed, we have added that the number of simulations with warm SST is not conclusive. The decision to not include SST as a parameter is discussed in an earlier comment about parameter choice.

**Line 342:** This subset of simulations shows that warmer initial SSTs may act to considerably speed up the transition, above meteorological conditions, which has implications for the future warmer climate. However, the PPE does not have enough simulations with warm SST to draw a definitive conclusion. The warm SST simulations have been removed from this analysis (leaving 34 simulations) since the difference in SST at initial stratocumulus is akin to perturbing a seventh parameter, but one that was not initially accounted for in our experimental design.

***Figure 5****: Add the number of simulations shown in each panel. In panel d, add the mean SST at T1 or T0 for each group to show how different they are. Also, why was the value of "296 K" chosen?*

Thank you for these suggestions, this figure (now Fig. 6) has been updated to include simulation numbers and mean SSTs at T1 for both subsets. Panel labels, legends and y-axis labels have been slightly altered too.

[Figure]

The three later simulations were split because they did not form stratocumulus for a considerable time after the others (over 12 hours). The value of 296 K is simply a threshold that was passed in that time, so it was a clean way to split them. It equally could have been based on T1 being over 24 hours in the simulation. We chose to frame in terms of the SST because the SST value is the factor that makes the difference in how the simulations unfold.

*Figure 6*: *The inset is missing. Also, what specific time do these points correspond to in each panel showing two parameters?*

Thank you for highlighting this text in the caption, there is not meant to be an inset for this figure (now Fig. 7). The sentence has been removed.

The points shown here correspond to new "input values" for the emulator. In the original draft of the manuscript that was reviewed the emulators were trained on the initial simulation values after spin up (mentioned in an earlier comment about regions of parameter space that form stratocumulus). However, on reviewing the manuscript we have become aware that

some of the parameter values shifted more between spin up and the formation of stratocumulus clouds. Since we are considering the transition time as beginning at T1 (formation of stratocumulus) we have now changed the input values to now correspond to the values at T1. This marginally affects the emulators and parameter space, and consequently Figs. 1, 7, 8, 10 and 11.

*L305–306: It would be helpful to add more explanation on the calculations and preparation of Figure 7. You previously stated that 1000 emulators were created. How do you have 1 million points here? It seems that Figure 7 involves more than just averaging the points from Figure 6, so more elaboration is needed.*

We apologise for the confusion in these figures. The emulator of transition time was used to generate 1000 new predictions for Fig. 6 (now 7) based on a Latin hypercube pattern. This fills the parameter space much more than the original PPE and the advantage of a Latin hypercube distribution for this figure is that it is the optimum distribution of points across 6 dimensions to enable the points to be seen reasonably well in each 2-dimensional panel. We included this because we feel it gives more credibility to Fig. 7 (8) where the emulator was used to predict another 1 million grid-based points that were then averaged through 4 dimensions in each panel. Comparing panels in Fig. 6 (7) and 7 (8) where the mosaic forms a pattern, to those where it does not, shows how clearly the stronger relationships in Fig. 7 (8) show through the 6 dimensions.

**Line 349:** The emulator's posterior mean response surface was used to make 1000 predictions of transition time, which fill the parameter space and provide far more information than the raw PPE data alone. These 1000 points are sampled from the emulator's posterior mean distribution using a Latin hypercube design, so each point varies in all 6 dimensions. Figure 7 immediately begins to inform us about the subtleties in variation across parameter space.

**Line 361:** The strength of the output's dependency on each parameter and the joint effects of parameters can be more easily interpreted  using an averaged response surface. Figure 8 shows 1 million grid-based points sampled from the emulator's posterior mean distribution and averaged through the 4 dimensions not shown in each 2-dimensional panel.

*L306–307: How is this quantified? Based on the inset in Figure 7, Na has the highest impact, followed by b_aut to a much lesser extent. Delta theta variance is very low, similar to BLz variance.*

The emulator set up has since been altered so the text in Section 3.3.2 has been rewritten. With the change in input values for the emulator, we explored again whether the emulator worked best with a linear mean function (as in the reviewed manuscript) or a constant mean function. We found that a constant mean function creates a better emulator this time. This is likely because there are larger gaps in the training data after some of the parameter values shifted before stratocumulus formed and the emulator mean function defaults to the mean function where it lacks training data. The linear mean function was adding too much of the linear function to the averaged response surfaces, whereas the constant mean function aligns better with the space-filling result in section 3.3.1.

See the "Updated figures" section at end of this document for the updated figures and accompanying main text.

**Figure 7 caption:** *Some methodology details seem to be missing, making the first sentence difficult to understand for readers unfamiliar with the method. Also, in the last sentence, do you mean "variance in transition time"?*

The caption has been changed as follows:

**Figure (8):** Averaged transition time response surface. The transition time emulator was sampled 1 million times using a 6-dimensional grid and a-o) shows each 2-dimensional combination of the six perturbed factors averaged through the remaining 4 dimensions not shown in that panel. The inset in the top right shows the contribution of each parameter's variance to the variance in the transition time.

**Figure 8:** *Panels k, l, and o are redundant; they are the same as in the previous figure and should be deleted.*

With the expansion of the paper and the results now focusing more on the PPE analysis, we have decided to remove the previous Fig. 8.

**Figure 9:** *Some tick labels are overlapping. The highest value on the y-axis (20) in panel c is hidden beneath the lowest value on the y-axis (0.0) in panel b. The "80" and "0" values in panels c and e appear as "800."*

Panel e shares a right-sided y-axis with what is now panel f. The panels have been moved further apart to avoid 80 and 0 looking like 800. The panel that showed R against transition time has been moved to Fig. 11, and we have added a panel showing the boundary layer aerosol split into the two subsets.

[Figure]

**Figure 9.** Ensemble timeseries split by mean rain water path. (a) The domain-averaged rain water path timeseries for each member split by temporal mean rain water path greater than 7 g m⁻² (blue) or less than (red). (b) The boundary layer accumulation mode aerosol aligned to T1 and coloured by mean R. (c) The cloud fraction timeseries as in Fig. 5c but coloured by mean R. (d) As in (c) but aligned to T1. The means over each subset (high or low mean R)

are shown in bold. (e) The number of data points used in calculating the mean of each subset at each timestep in (c). f) As in (c) but for (b) and (d).

See the "Updated figures" section at the end of the document for main text changes about Fig. 9.

***L378–380***: *The length scale of open-cell stratocumulus clouds is usually greater than 10 km, so I do not think your LES setup could simulate them.*

We agree and have removed mentions of open-cell stratocumulus clouds.

***L384–385***: *Based on Figure 10, BL Na and delta theta are also important factors. Specifically, BL Na (for all cases and for those with higher R) and delta theta (for cases with lower R) have higher correlations with transition time. This should be mentioned in the text.*

The main text describing Fig. 9f and Fig. 10 have been rewritten (see "Updated figures" section).

***Figure 10***: *I recommend moving panel f from Figure 9 to Figure 10. That would make all panels in each figure consistent and avoid showing panel f before panel d in Figure 9. It would also help justify having Figure 10 in your paper. Currently, its explanation in the text is less than two lines.*

Thank you for this suggestion. We agree that it allows more consistency in the figures.

[Figure]

**Figure 10.** One-dimensional scatter plots of (a) $\Delta\theta$, (b) BL Na and (c) mean R against transition time. The scatter points show the 34 simulations that transitioned within the simulation time and are coloured by high mean R (blue circles) or low mean R (red triangles). Lines of best fit Pearson's correlation coefficients (r) and p values (p) are calculated for the whole set (black) and each subset.

***L386–387***: *Are the correlation values mentioned here and in Figure 9f statistically significant? Given the small number of simulations and low correlation values in some categories, some of these may not be significant. It would be good to calculate and report statistical significance here.*

All the correlation values have now been updated with statistical significance. The correlations that we were ascribing importance to have all shown to be significant. The

weaker correlations are not shown to be significant. See the previous comment for the updated figure.

*L431–437: The issue is not limited to autoconversion. There are various assumptions and tuning parameters throughout a microphysics parameterization. That's why intercomparison studies often show a wide range across different models and schemes.*

Correct, this is the point we are making in this paragraph. Thank you for re-stating it.

*L450–455: Can you elaborate on how this parameterization limitation might bias the drizzle-depletion mechanism or other processes? In particular, I think the shallow boundary layer in your case delays the transition time in the baseline and other simulations.*

Yes, that is correct. We have added this information into the text.

**Line 504:** A shallower boundary layer throughout the ensemble will likely delay the transition time in all simulations.

*L457: If your microphysics include the coarse mode, it should be mentioned in the methodology section where you describe Aitken and accumulation modes.*

It is mentioned in the methodology on line 138 that the aerosol can grow into the coarse mode.

*L460–461: There is no figure or explanation about the impact of different aerosol modes on the results. If you performed sensitivity tests, please include them in the supplementary material, along with an explanation in the Results section.*

We did not do specific sensitivity tests analysing the impact of different aerosol modes on the results. When first analysing the simulations we looked at the concentrations of the aerosol modes throughout the simulations. This can now be seen in Fig. 4d.

*L465–472: This is an important challenge in defining transition time. Another contributing factor is the diurnal cycle: it is possible that some cases labeled as "transition" simply reflect this cycle. This caveat can be added here.*

Yes, the recovery with the diurnal cycle is what we are talking about in line 467. We have made this more explicit:

**Line 517:** As mentioned previously, some of the cumulus clouds may have recovered to stratocumulus after the simulation ended, as part of the diurnal cycle. Similarly for the clouds that began with stratocumulus, there is an unquantifiable amount of time before the simulation where the cloud may have been formed.

*L465–472: Related to the previous point, you should mention here that the definition of transition time is based on T1 and T2, which themselves are based on fc > 0.9 and fc < 0.55, respectively. These values seem somewhat subjective or arbitrary, so varying them might change some results (unless you've done sensitivity tests).*

See comment on the sensitivity test above.

*L474–478*: As I wrote earlier and you correctly acknowledged here, the length scale of open-cell stratocumulus clouds is usually larger than the domain size used in your study and in the reference you cite. I reviewed that reference, and although they mentioned open cells in the abstract and other sections, they do not show open-cell morphology. In fact, they note in the middle of the paper that their domain is too small to resolve open cells. So, to avoid confusion and be more accurate, it is best to remove any conclusions about open cells.

We take this point on board and have removed any mention of open cells.

Reviewer 2:

*An ensemble of large eddy simulations of the stratocumulus to cumulus transition are performed, building off the Sandu and Stevens (2011) case study with variations in initial conditions (boundary layer depth, moisture and aerosol concentration, the jumps in moisture and potential temperature across the inversion) and microphysics (the dependence of autoconversion on the cloud droplet number). The joint variation of these parameters is chosen using a latin hypercube sampling with the goal of maximizing the minimum distance between ensemble members in this high-dimensional space. A brief inspection of the behavior of this ensemble is made before the focus shifts to training the Gaussian process emulator, which "interpolates" properties of the ensemble in the high-dimensional space to show their parametric dependence more clearly. The analysis of the emulator focused on how the timing of transition and the mean rain water path depend on the various parameters, with the strongest dependence on initial aerosol, then inversion stability and auto conversion for the transition time.*

*Recommendation: Major revisions*

*The paper is well written and tells a nice story about the transition. As a person who has made simulations like those in the ensemble, I wish the authors had shared more about the results of those simulations before shifting to the emulator results. If there's another manuscript being prepared about those simulations, the authors could highlight that forthcoming manuscript in the paper but might still consider including a bit more in this paper. The paper would also benefit from more interrogation into the drizzle-depletion vs. deepening decoupling transitions. I'll make some suggestions along these lines in the major comments below. Several of my suggestions would involve a fair amount of effort on the part of the authors, so I would understand if they chose not to pursue all of them.*

=========================

*Major comments (11/240 means p. 11, line 240):*

***A tremendous amount of computational effort was put into the ensemble of LES simulations, but the paper passes quickly over the actual simulations and spends more time talking about the smoothed/filtered/interpolated view of the simulation results presented by the emulator. It would make sense to have a couple of figures (possibly in a supplement) that summarize the simulation results. If the authors had some set of plots that they used to understand the broad behavior of the simulations, those would work well for this purpose. If the authors don't have something like that, my suggestion would be a collection of time series: some subset of SST, inversion height, lifting condensation level, decoupling, accumulated precipitation, boundary layer aerosol concentration, cloud fraction, liquid water path, rain water path, shortwave cloud radiative effect. Depending on the behavior of the runs, a presentation following left two panels in figure 3 of Chen et al (2024, https://doi.org/10.5194/acp-24-12661-2024) that contrasted the behavior of the drizzle-depletion, deepening-decoupling and no transition group of the simulations, might make the plots easier to decipher. One or two phase space plots such as Figure 1 of Glassmeier et al, 2019 (https://doi.org/10.5194/acp-19-10191-2019) could also give the***

*readers some idea of how the simulations are similar to and different from each other according to the different metrics.*

We have added a figure that summaries the PPE according to the suggested categories in Section 3.2 PPE Summary.

[Figure]

**Figure 5:** Summary of the whole PPE. a) Sea-surface temperature forcings applied to all simulations, b) temperature inversion height, c) lifting condensation level, d) decoupling factor based on , e) accumulated surface precipitation, f) cloud fraction, g) liquid water path, h) rain water path, and i) boundary layer accumulation mode number concentration. The PPE is split into three categories 1) members that formed stratocumulus but did not transition, 2) members that transitioned but had a mean rain water path of less than 7 g m^(−2), and 3) members that transitioned but had a mean rain water path of more than 7 g m^(−2). The line shows the median of each subset and the shading shows the minimum and maximum of the subset. The grey shading indicates local nighttime.

*I would also suggest a brief exploration of the contrast between simulations that exhibit drizzle-depletion and deepening-decoupling transitions (maybe resulting in one extra figure if the results are interesting to the authors). A couple of possible questions: Does the boundary layer aerosol actually decrease in the drizzle-depletion simulations? Does boundary layer aerosol decrease more than in the drizzle-depletion transitions than in deepening-warming ones, or is the initially low aerosol in the drizzle-depletion more important?*

We have adapted Fig. 9 to now include the timeseries of boundary layer accumulation mode aerosol for both subsets of the ensemble. Figure 9f has been moved to Fig. 10. In most simulations the boundary layer aerosol does indeed decrease through the simulation. The high mean R subset has a median concentration that is initially higher than the low mean R

subset, but it decreases more sharply over the first 20 or so hours from T1. After 20 hours, the gradient of the high mean R subset levels out to be similar to the low mean R subset.

[Figure]

*Framing the analysis around the influence of aerosol on the timing of the transition is well chosen.  However, the transition is of interest primarily because the associated changes in cloud cover likely cause significant changes to the radiative balance at top of atmosphere (TOA).  Might the authors take a look at how much of the difference in shortwave cloud radiative effect (CRE) or TOA net shortwave flux across the simulations is explained by differences in transition time?  (A consideration of longwave fluxes could also be included but would probably have weaker signals across the transition.)  Here, I recognize that --- if the transition occurred during nighttime hours --- the shortwave radiative fluxes might not vary much if the runs transitioned at different times when the sun was down.*

This would be an interesting piece of analysis to do however we think it is beyond the scope of this manuscript.

*Regarding sec 2.3: Would the transition threshold be less noisy/sensitive if the transition metric incorporated some time averaging of cloud fraction (i.e., three hour average f_c < a threshold value) or, alternatively, required that f_c remain below a threshold value for a few consecutive hours?  Might this narrow the emulator uncertainty as represented by the length of the vertical lines in Figure 3a?*

If we could continue the simulations for longer, either of these options would have been a nice addition. We tried requiring the f_c remain below the threshold for 12 hours when deciding on a good metric of transition time. This resulted in only 12 simulations that transitioned within the simulation. Whilst the correlations were still apparent, they were no longer statistically significant. Similarly, with setting a rolling average we also lose a significant number of simulations at the end, which is when the transition tends to occur. We have opted to continue with our metric of transition time, but we have also now done a sensitivity test to show that the correlations hold for a lower threshold too. See the response to reviewer 1 for details.

=========================

Specific/minor comments

*2/30-32: I would be inclined to reference Bretherton and Wyant (1997) at the end of this sentence.*

Agreed and added.

**Line 35:** As the boundary layer deepens, mixing throughout the full layer can no longer be sustained and the layer decouples into a stratocumulus cloud layer and a surface-coupled sub-cloud layer (Bretherton and Wyant, 1997).

*2/32-33: Re-wording suggestion: "Once decoupled, the moisture is supplied to the stratocumulus by cumulus plumes emerging from sub-cloud layer, rather than eddies driven by cloud-top radiative cooling." Is there really an interval where the stratocumulus moisture supply is "cut off"? This might be true in warm advection cases when cold SSTs would lead to negative buoyancy fluxes at the surface and near-surface stable layers, but, in the present transition simulations where the SST increases with time, I would have thought that cumulus should be supplying moisture to the stratocumulus layer from the onset of decoupling. With a small domain, there might be some intermittency in the cumulus occurrence, but I would be surprised if the horizontally-averaged total water flux at the top of the subcloud layer systematically falls to zero at the onset of decoupling in the MONC simulations studied here. Does that actually occur? One other point, the "warmer" sub-cloud layer referenced on line 33 is coming from the increase in SST over time. The process of decoupling leads a cooling and moistening of the sub cloud layer relative to the cloud layer.*

Thank you for the suggestion and pointing out the mistake on line 33. We agree that the re-worded sentence is a more accurate description and have changed the text accordingly. We have also clarified that the sub-cloud layer is not warming relative to the cloud layer.

**Line 31**: It describes how increasing SSTs cause the boundary layer turbulence to be increasingly driven by surface fluxes that deepen the boundary layer and enhance the entrainment of warm and dry air at cloud top. As the boundary layer deepens, mixing throughout the full layer can no longer be sustained as the sub-cloud air cools and moistens, and the boundary layer decouples into a stratocumulus cloud layer and a surface-coupled sub-cloud layer (Bretherton and Wyant, 1997). Once decoupled, the moisture is supplied to the stratocumulus by cumulus plumes emerging from the sub-cloud layer, rather than eddies driven by cloud-top radiative cooling. In this cumulus-under-stratocumulus stage, the plumes at first provide moisture and turbulence to the stratocumulus layer, but more-energetic plumes overshoot and vigorous mixing eventually dissipates the stratocumulus cloud resulting in a field of cumulus.

*2/46-48: I would suggest adding a phrase/sentence to set the reader's expectations, something like, "While increases in aerosol or cloud droplet number concentrations might be expected to delay the transition due to precipitation suppression, Chun et al. (2025) ..." The work of Ackerman et al (2004, https://doi.org/10.1038/nature03174) has some relevance to this, though I'm not sure whether a citation is needed here.*

These are both good suggestions. The sentence on Chun et al. has been reworded to better fit the point and remove the "detailed microphysics scheme" wording.

**Line 48:** However, as in many LES studies, a fixed droplet number was used, while Yamaguchi et al. (2017) showed that aerosol collision-coalescence processes are required to represent droplet depletion. Chun et al. (2025) included aerosol processing and found that aerosol injection suppressed precipitation, however they found the aerosol effect on the transition is overestimated where large-scale circulation adjustments are ignored.

*5/126: Since precipitation is so important to the transition in at least some of the cases, I would encourage the authors to mention that precipitation onset tends to occur sooner in larger domains. Yamaguchi et al (2017, Fig. 1) show this in transition simulations based on Sandu and Stevens (2011) and also include as nice discussion about this in the middle paragraphs on p. 2345. Efrani et al (2022, Figure 6) show the effect of domain size on precipitation onset and transition timing in a case study with prognostic aerosol.*

Thanks for this suggestion, the following sentence has been added:

**Line 149:** The horizontal resolution was 50 m, and the vertical resolution varied from 20 m near the surface, to 5 m around the temperature inversion, and gradually increased above that. It is worth noting that the domain size affects precipitation formation, with precipitation onset occurring earlier in larger domains where mesoscale organisation can be simulated. Yamaguchi et al. (2017) show sensitivity tests for different domain sizes, and Erfani et al. (2022) found that a large domain size encouraged earlier precipitation and onset of the stratocumulus-to-cumulus transition. The LES setup is idealised because realistic profiles would be specific to an individual transition case rather than being representative of a typical case. Although this may limit the realistic nature of the simulations, it simplifies the perturbation method for a study such as this where perturbations are made from a reference case to learn broadly about the transition behaviour across parameter space.

*If the computational effort isn't prohibitive, could a couple of sample simulations (perhaps one drizzle-depletion and one deepening-decoupling simulation each) be simulated in a larger ~25 (or better yet ~50) km square domain. It would be supportive of the present results if both simulations showed the same type of transition in the larger domain even if the time of transition changed. Even if not, it could be mentioned as a qualification. Making all of the simulations in a larger domain would be much more expensive, so there's no expectation here that they should all be redone in a larger domain.*

Unfortunately, we are not able to run any further simulations for this research. This would be a great test to do though.

*6/166-7: Please specify the free tropospheric aerosol concentration for both Aitken and accumulation mode aerosol. Because of entrainment, the boundary layer aerosol concentration would tend towards the free tropospheric value over time in the absence of any other sources and sinks. If the free tropospheric aerosol concentration was smaller than the initial value in the boundary layer, this would make precipitation more likely towards the end of the simulation.*

The free tropospheric values have been added in the "Boundary layer aerosol" section at line 194. The FT accumulation mode concentration is fixed for all simulations, but the BL accumulation concentration varies, so the simulations vary in whether the FT is more or less than the BL. We will leave this comment out of the manuscript.

**Line 194:** The initial boundary layer concentration of accumulation mode aerosol was perturbed because the vast majority of aerosols that activate into cloud droplets (cloud-condensation nuclei) are from the accumulation mode. Boundary layer Aitken mode was initialised with a concentration of 150 cm^(-3) and allowed to freely evolve. Free-tropospheric aerosol can also be a source of cloud-condensation nuclei and could be important in simulations with very low aerosol concentrations in the boundary layer (Wyant et al., 2022). However, free-tropospheric aerosol concentration was kept constant across the PPE because it was not expected to be as important as the key factors chosen. The Aitken concentration was 200 cm^(-3) and the accumulation concentration was 100 cm^(-3). There is no surface source of aerosol throughout the simulations.

*6/171: Why not replace -1.79 with b_{aut} in this equation? Then, modify the following sentence towards the end: "... (both in kg kg^-1), N_d is the cloud droplet number concentration (cm^-3), and b_aut is the exponent of cloud droplet number concentration, which has the value of -1.79 in Khairoutdinov and Kogan (2000). We perturb the value of b_aut to change the autoconversion rate, and this is one of the parameters varied in our PPE."*

Thank you, this is a good suggestion.

**Line 205**: … (both in kg kg−1), and Nd is the cloud droplet number concentration (cm−3), and b_aut is a model parameter. We perturbed b_aut from the default value of -1.79 to perturb the autoconversion rate. The default parameter values were estimated in Khairoutdinov and Kogan (2000) by reducing the mean squared error between the above function and an explicit microphysics model, and there are large uncertainties surrounding each of these values.

*11/256-264: The diurnal cycle plays a role in these transition simulations, so it would be useful to note the local time of day of these snapshots in the text and/or caption.*

The time of day is now added in the figure caption and each snapshot is now from 9pm, rather than beginning, middle and end.

*Figure 4 suggestions:*

*- Add symbols in cloud fraction (and perhaps other panels at left) showing times of the panels (a), (b) and (c).*

*- Use a log colorscale for liquid water path. The contrast between 20 and 40 g/m2 is arguably more important than the one between 200 and 400 g/m2.*

*- Add a timeseries panel at left showing the boundary-layer-mean aerosol concentration.*

Thank you for these great suggestions, we have incorporated them all into Fig. 4. We have also included the inversion height and the lower threshold for the masked values on the cross sections, as suggested by reviewer 1.

[Figure]

**Figure 4.** Base simulation cloud properties. a-d) timeseries of cloud fraction (f_c), liquid water path (L), rain water path (R), and boundary layer aerosol concentrations (N_a). Grey shading indicates local nighttime. e-j) Snapshots at 9pm local time on day 1 (e-f), day 2 (g-h) and day 3 (i-j). Top row (e, g, i) shows top-down views of L and bottom row (f, h, j) shows vertical cross sections of liquid water mass-mixing ratio (MMR) at the y-location of the transect line. The MMR is masked for values lower than 0.01 g kg^{-1}, in line with the f_c definition.

*16/330-332: My reading of Figure 7i is that the time to transition grows shorter with a faster autoconversion timescale for both shallow and deep boundary layers. It's not clear to me that precipitation is causing the transition to be delayed in shallow boundary layers.*

Thanks for this comment. Perhaps some of the figure description was stretching the available information. The figure analysis has been rewritten in light of the new emulator that was produced. See the updated figure and text in the "Updated figures" section at the end of this document.

*As noted above (2/32-33), the language about the stratocumulus being "cut off" from its moisture source seems too strong in my mind.*

This phrasing has been removed.

**16/348-349:** Regarding "Moist boundary layers allow thicker clouds to form, which would then take longer to dissipate through entrainment", wouldn't thicker clouds also be more likely to precipitate and reduce boundary layer aerosol through collision-coalescence scavenging? It's not obvious to me that thicker clouds would delay transitions in all scenarios.

Yes, this is also true. In these paragraphs we were suggesting potential mechanisms that might lead to these relationships, but the very nature of the problem is that there are confounding effects like these, and it is hard to know which wins out. This figure has since been updated and we no longer talk about the specific humidity relationships.

*18/Fig. 8: Might fewer but larger panels at left tell the story as well or better? That might allow the smaller panels at right to be larger, especially if the transition time colorbar was*

*made vertical, so that those panels could be as large as the others. The changes in the bottom row of panels at left seem much more gradual than the top row to my eye, so that dropping a couple of those might make the panel-to-panel changes more striking.*

*Also, label the panels at left so they can be individually reference in the text. More broadly, labeling all of the sub panels with each figure would be helpful.*

With the expansion of the paper and the results now focusing more on the PPE analysis, we have decided to remove the previous Fig. 8. We have now added panel labels to all figures.

**18/360:** *Regarding "For BLNa <100 cm−3, the transition time is very low and almost invariant to the other two parameters.", the transition time seems to vary by ~40 hours with changes in delta theta along the top edge of panel 8b. If this sentence was referring to something else, maybe specify the figure panel explicitly for clarity.*

This figure has now been removed.

**19/Fig. 9:** *Might the maximum rain water path or the accumulated surface precipitation or some other metric divide the ensemble into subgroups with less overlap in the cloud fraction evolution? Did the authors try other metrics and find that they behaved similarly?*

Yes, we tried a few different rain water path metrics and did not find much of a difference.

**19/378:** *If I understand things correctly, this behavior seems similar to that seen in the fixed-Nd simulations of Sandu et al (2008, https://doi.org/10.1175/2008JAS2451.1).*

This behaviour could be along the same lines. Sandu et al 2008 find that the pristine case, which has more precipitation, recovers to almost the same LWP values through the night, whereas the polluted case, with suppressed precipitation, never reaches the same LWP values as the first night. In our figure, we see that the mean $f_c$ for higher precipitation cases recovers more through the night compared with the mean $f_c$ for lower precipitation cases, which have a more steady and consistent decline. However, in these $f_c$ means we see that the higher precipitation mean also has more extreme decreases through the day, whereas in Sandu et al. 2008 the overall diurnal cycle is dampened for the drizzling case. We have mentioned the reference in the text.

**Line 416:** In Fig. 9c), the timeseries are lined up with the diurnal cycle and it shows that the high R subset mean recovers more than the low R mean during the nights. This could be similar to the behaviour shown in Sandu et al. (2008), where the drizzling stratocumulus case recovers to higher L values through the night compared with the suppressed precipitation case, which is driven more by entrainment than longwave cooling. However, Figure 9 indicates that some of the high R cases follow a more extreme diurnal cycle than the low R cases. This might suggest that the simulations with more initial rain transition to a state like open-cell stratocumulus rather than cumulus, which may show larger fluctuations. However, the domain in our simulations is likely too small to simulate open-cell stratocumulus. In Diamond et al. (2022), they found that drizzle depletion caused the stratocumulus to transition to open-cell behaviour rather than cumulus, but did not determine which factors would cause one transition over the other.

***22/421-423:*** *Could the "ultra-clean layers" observed during CSET be one stage (or the remnants) of such a transition?*

Yes, this is a good point. We have added a couple of sentences about this.

**Line 461:** The PPE approach also reveals that the part of parameter space with a particularly strong aerosol effect is small, which could explain why fast transitions by drizzle depletion in the real world have not been observed very well. It is unlikely that campaigns, particularly in the NE Pacific Ocean off the coast of North America, will observe conditions of particularly deep, pristine boundary layers, hence there are no clear observations of a low-aerosol induced rain-hastened mechanism in this region. However, "ultra-clean layers" where the concentration of particles larger than 0.1 um is below 10 cm^(-3), are a common feature of the transition and may be the result of the drizzle-depletion mechanism (Wood 2018, O 2018).

***23/439-440:*** *It might have been nice to hear more about the tendency of the model to produce less drizzle earlier in the paper.*

This is in reference to the base case producing less drizzle, which is discussed in Section 3.1 (line 266 - 274). We have now made the reference to Section 3.1 clear.

**Line 487:** The drizzle-depletion effect is weaker in our simulations, which may be due to our model producing less drizzle (seen in the base case in Section 3.1) and also because many of our simulations form drizzle much earlier, with peaks in the first or second day.

***23/457:*** *In addition to Wyant et al (2022), McCoy et al (2024, https://doi.org/10.1175/2008JAS2451.1) also looked at Aitken buffering in simulations of a case over the Northeast Atlantic Ocean.*

Thank you for the suggestion, the reference has been added. We have also added Merikanto et al., (2009).

**Line 507:** Merikanto et al., (2009) first showed that a significant portion of marine boundary layer cloud-condensation nuclei are formed in the free troposphere. More recently, the Aitken buffering hypothesis of McCoy et al. (2021) has been supported by simulations in Wyant et al. (2022) and McCoy et al. (2024) that show Aitken-sized aerosol can be transported to the boundary layer where the larger particles act as cloud condensation nuclei.

=========================

*Typographical/wording suggestions (Optional):*

***0.*** *Does "boundary layer aerosol concentration" include both accumulation and Aitken mode aerosol? If so, how are they partitioned? If not, how do the two change over time? Maybe include time series of both in the suggested plots in major comment 1 above.*

We are not quite sure where in the paper this is referring to, but perhaps the paragraph on the perturbed aerosol. This paragraph has been expanded now (see above) to describe the Aitken and accumulation concentrations better. Only the accumulation mode was perturbed.

Figure 4 now includes timeseries of both Aitken and accumulation and Fig. 9 has the ensemble timeseries too.

*2/22: "reduce" --> "weaken". I think the "weaken"/"amplify" contrast might work better.*

Good suggestion, this is incorporated in the rewriting of this sentence in a previous comment.

*4/97: "... also varies the dependence of cloud-to-rain autoconversion on the cloud droplet number concentration"*

Agreed and added.

**Line 101:** Given the potential importance of drizzle formation, the ensemble also varies the dependence of cloud-to-rain autoconversion on the cloud droplet number concentration.

*11/254: Insert "(see also, " before citations in parenthesis.*

Agreed and changed.

*14/307: Left parenthesis before delta theta.*

Thanks for spotting this. Changed.

*16/all: Maybe include the figure number each time the panels are referenced?  I had to look back a couple of pages to find the figure number.*

This is a good point. We have added more references to the whole figures.

*16/348: "longer transitions" --> "later transitions"*

Agreed and changed.

*18/367: Maybe add "(R)" after rain water path in the section title, as a reminder to the reader.*

Agreed and changed.

[revised manuscript text omitted]

---

## Author Response (AR2)

**Response to Reviewers**

Manuscript title: Strong control of the stratocumulus-to-cumulus transition time by aerosol: analysis of the joint roles of several cloud-controlling factors using Gaussian process emulation

Corresponding author: Rachel W. N. Sansom

We thank this reviewer again for dedicating time to improving this manuscript to make it publishable. We are very grateful for the time invested and we are excited to get it published. Personally, I absolutely will take on board their comments about writing the response to reviewers more succinctly. It was a big effort to write and, again, I really appreciate that you took the time to read it.

The first four points from the reviewer have been corrected as suggested:
2/39: "... driven by cloud top cooling _and surface fluxes._" I think the "cloud top cooling" was my suggestion, so I wanted to mention the surface fluxes, since they do play a role as well.

6/167: I think the exponent here should be "-6" rather than "-26".

9/270: "to be significant" seems a bit vague here. Maybe "... to give statistically significant information about the relationship of cloud controlling factors and transition time." or something like that.

13/342: "... due to the different _numerical methods,_ radiation schemes and mixing processes ..."

14/357: Is "cloud base" the stratocumulus cloud base, the median/mean cloud base height, or something else? Apologies if you've defined it already and I missed it.
The following text has been added to clarify.

**Line 323:** Cloud base was calculated as the domain-mean cloud base where cloud is present in the column.

30/648: Missing journal in Bretherton (2015) reference.
This reference is present, but the ACP reference style splits citations by single author first, then co-authored, and then team authored. For Bretherton, we have cited enough papers that all three categories are included. The 2015 paper is cited first, rather than chronologically for all Bretherton references.